# Biosynthetic energy cost for amino acids decreases in cancer evolution

Hong Zhang[1], Yirong Wang [1,2], Jun Li [3], Han Chen[3,4], Xionglei He[4], Huiwen Zhang[5], Han Liang [3,6] & Jian Lu [1]

Rapidly proliferating cancer cells have much higher demand for proteinogenic amino acids than normal cells. The use of amino acids in human proteomes is largely affected by their bioavailability, which is constrained by the biosynthetic energy cost in living organisms. Conceptually distinct from gene-based analyses, we introduce the energy cost per amino acid (ECPA) to quantitatively characterize the use of 20 amino acids during protein synthesis in human cells. By analyzing gene expression data from The Cancer Genome Atlas, we find that cancer cells evolve to utilize amino acids more economically by optimizing gene expression profile and ECPA shows robust prognostic power across many cancer types. We further validate this pattern in an experimental evolution of xenograft tumors. Our ECPA analysis reveals a common principle during cancer evolution.

[1] State Key Laboratory of Protein and Plant Gene Research, Center for Bioinformatics, School of Life Sciences and Peking-Tsinghua Center for Life Sciences, Peking University, Beijing 100871, China. [2] Academy for Advanced Interdisciplinary Studies, Peking University, Beijing 100871, China. [3] Department of Bioinformatics and Computational Biology, The University of Texas MD Anderson Cancer Center, Houston, TX 77030, USA. [4] State Key Laboratory of Biocontrol, School of Life Sciences, Sun Yat-sen University, Guangzhou 510275, China. [5] Department of Biochemistry and Molecular Biology, The University of Texas Health Science Center at Houston, Houston, TX 77030, USA. [6] Department of Systems Biology, The University of Texas MD Anderson Cancer Center, Houston, TX 77030, USA. These authors contributed equally: Hong Zhang and Yirong Wang. Correspondence and requests for materials should be addressed to H.L. (email: hliang1@mdanderson.org) or to J.L. (email: luj@pku.edu.cn)

Cancer development is a multiple-step evolutionary process in which cancer cells acquire a selective advantage in their competition with neighboring cells[1–3]. With the advancement of high-throughput genomic characterization technologies, extensive studies have systematically elucidated the molecular basis of human cancers[4,5]. One striking observation is the tremendous diversity of distinct molecular mechanisms among different cancer types, among different samples of the same cancer type, and even within a single tumor[6]. Regardless of the specific molecular changes occurring in each cancer, cancer cells must adapt to their microenvironment for rapid proliferation[6,7] and metabolic adaptation is the key to this process[8,9]. Indeed, metabolic reprogramming has been proposed as a hallmark of cancer cells[6,7,10]. However, quantitative characterization of metabolic adaptation at the cellular level remains challenging.

Amino acids (AAs), the building blocks of proteins, are an essential class of metabolites. As the composition of cellular biomass is dominated by proteins[11], the regulation of protein synthesis and AA usage is particularly important for cancer cells, which have an enhanced demand for AAs to support their rapid growth[12,13]. Mammalian cells can endogenously synthesize only 11 AAs, known as nonessential AAs (NEAAs)[14] and have to obtain the remaining 9 AAs, known as essential AAs (EAAs), from the diet[15] or microbes[16]. However, the endogenous synthesis of NEAAs might not be sufficient for the proliferation of cancer cells, as the reduced exogenous supply of NEAAs such as glutamine can impair the survival or tumorigenic potential of malignant cells[10,17–19]. Importantly, recent metabolic profiling experiments have demonstrated that cancer cells obtain EAAs and some NEAAs from external sources for protein synthesis[11,20]. Despite the importance of AAs to the proliferation of tumor cells, it remains unclear how the usage of AAs in protein synthesis affects cancer progression.

The use of different AAs in proteomes is presumably constrained by their biosynthetic energy cost, which varies greatly regarding the high-energy phosphate bonds consumed in biosynthesis in living organisms. In autotrophs (bacteria, yeast, and plants), which can synthesize all 20 proteinogenic AAs, biosynthetically inexpensive AAs are preferentially utilized over "expensive" AAs in the proteomes[21–26]. The anticorrelation between the biosynthetic cost and usage (termed C–U anticorrelation hereafter) of AAs appears to be driven by natural selection for bioenergetic efficiency in the autotrophs[22]. Intriguingly, although animals can synthesize only 11 NEAAs[14], significant C–U anticorrelations have been observed for all 20 AAs in humans and other animals when the cost of AA biosynthesis in bacteria[23,24,27] or yeast[28] is employed. A reasonable explanation is that EAAs and most NEAAs in animal cells are ultimately taken from the autotrophs in which the bioavailability of an AA is constrained by its biosynthetic cost[23,24,28]. Based on this hypothesis, the biosynthetic cost of AAs, combined with gene expression profiles, should well reflect how cells manage the expenditure of all 20 AAs in protein synthesis.

In this study, we introduce the concept of energy cost per AA for a gene ($ECPA_{gene}$) to measure the average biosynthetic cost of AAs in a gene/protein. Based on $ECPA_{gene}$ and the overall gene expression profile of a sample, we calculate $ECPA_{cell}$, which is a quantitative index for the average biosynthetic cost of AAs in the proteomes of the cells. As the EAAs and most NEAAs in human cells are ultimately taken from the autotrophs, neither $ECPA_{gene}$ nor $ECPA_{cell}$ measures the actual energy human cells invest to synthesize the AAs endogenously. Instead, these parameters can be treated as the average price tag for the AAs in a protein or the proteome, respectively. Therefore, lower ECPA values indicate reduced relative usage of expensive AAs and vice versa. Using these two parameters, we investigate how cancer cells evolve to utilize AAs more economically by optimizing gene expression profiles.

## Results

### The biosynthetic cost underlies AA usage in human proteomes.
Previous studies have demonstrated the C–U anticorrelation in a limited number of species (108 genomes[23] and 43 genomes[24]). To test whether this is a general pattern, we examined the relationship between the biosynthetic cost and usage of AAs in 11,253 species spanning bacteria, archaea, protists, plants, fungi, invertebrates, and vertebrates (see Methods). Taking humans as an example, we counted the number of each AA in all the protein sequences (Fig. 1a) and conducted a correlation analysis between the occurrence ($log_2$) of AAs and the biosynthetic cost (Supplementary Table 1) that was normalized by the AA decay rate as previously described[23] (Fig. 1b). As expected[23,24,28], we detected significant C–U anticorrelation using the biosynthetic costs of AAs in bacteria (B20, Pearson's $r = -0.89$, $P = 1.3 \times 10^{-7}$) or yeast (Y20, $r = -0.89$, $P = 1.8 \times 10^{-7}$) (Fig. 1b). Our analyses in other species reveal that Pearson's $r$ ranges from $-0.95$ to $-0.5$ ($P < 0.05$ in $> 99\%$ of the species), with a median value $< -0.8$, suggesting that the C–U anticorrelation is universal across all seven clades. As the AA biosynthetic cost is highly conserved between bacteria (B20) and yeast (Y20, Supplementary Table 1), in each species, the analyses with B20 and Y20 yielded nearly the same results (Fig.1b–d).

Next, we questioned whether the C–U anticorrelation existed if we focused only on the 11 NEAAs that can be endogenously synthesized in human cells. As the biosynthetic pathway of NEAAs might be different in humans compared with yeast or bacteria[29], we calculated the biosynthetic cost for each NEAA in humans (H11) following previous studies in bacteria[22] or yeast[21,26], while taking into account the differences (Supplementary Methods, Supplementary Table 1 and Supplementary Figures 1–3). We see that the relative costs for NEAAs are very similar among humans, bacteria, and yeast (Fig. 1e), and still observe significant C–U anticorrelations in humans (Fig. 1b) and other animals (Fig. 1c) with the H11 metric. It is not surprising that the correlations obtained with H11 are weaker than those obtained with B20 or Y20 (Fig. 1c, f), as only 11 AAs were used in the analyses. We further confirmed the C–U anticorrelations in humans and five other species with permutation tests by randomly shuffling the cost (B20, Y20, or H11) of AAs 10,000 times and conducting correlation analysis (Supplementary Fig. 4). Taken together, the universal C–U anticorrelation suggests that the biosynthetic cost underlies the usage of AAs not only in autotrophs but also in heterotrophs, such as humans.

Despite its prevalence, the C–U anticorrelation has not been verified with experimental data in autotrophs nor in heterotrophs. Herein, we provide evidence that the abundance of AAs hydrolyzed from proteomes of bacteria[30] or yeast[31] is significantly anticorrelated with the B20 or Y20 cost metric, respectively (Supplementary Fig. 5a, 5b). Moreover, the abundances of AAs hydrolyzed from proteins in whole bodies of rats, sheep, pigs, and chickens[32] show significant anticorrelations with the biosynthetic cost of all 20 AAs (B20 or Y20) or the 11 NEAAs (H11) (Pearson's $r \leq -0.63$, $P < 0.05$ in each test; Supplementary Fig. 5c). Our permutation analysis (Methods) further confirmed these patterns (Supplementary Table 2). To our knowledge, we provide the first experimental evidence that the biosynthetic cost governs the composition of AAs in proteomes of autotrophs and animals.

Human intracellular AAs come from two sources: (1) NEAAs endogenously synthesized in human cells or other animal cells

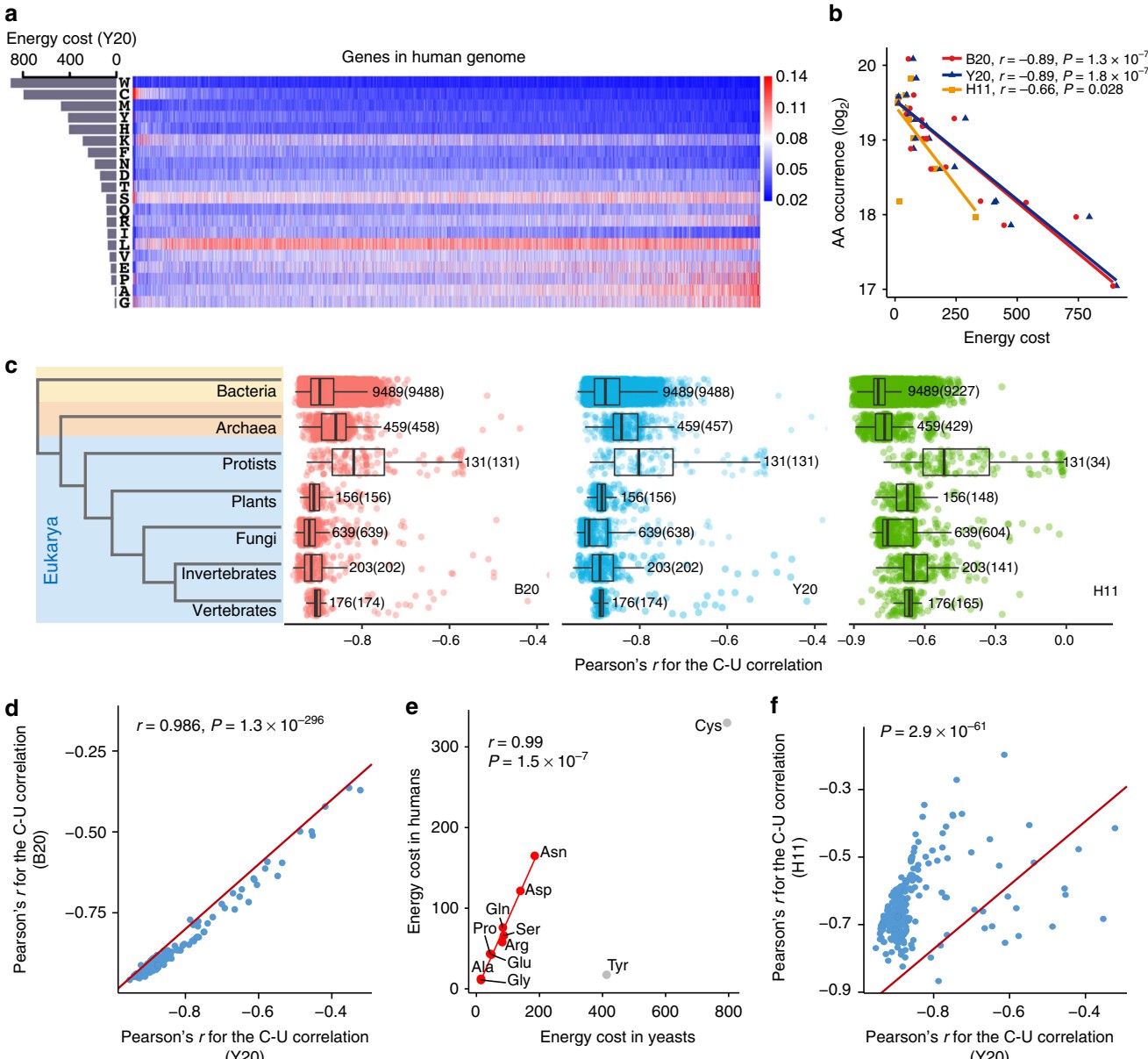

**Fig. 1** Biosynthetic cost of AAs is correlated with AA usage in protein sequences. **a** Proportions of 20 AAs in human proteins. Bar plot on the left shows the biosynthetic cost of each AA (Y20). **b** The relationship between AA occurrences ($\log_2$) in all human protein sequences and cost of AAs (red point, blue triangle and green square for B20, Y20, and H11, respectively). Pearson's correlation test was performed. **c** Boxplots showing the distribution of Pearson's $r$ for the C–U correlation in seven major taxonomic groups in all domains of life. Phylogenetic tree at left shows the evolutionary relationship between the seven groups. The number of species in each of the seven groups is presented and the number of species showing significant C–U anticorrelation ($P < 0.05$) is given in parentheses. Due to the conservation of cost metric or food chain, significant C–U anticorrelation was observed in all domains of life with three cost metrics (B20, Y20, and H11). Center line, median; box limits, upper and lower quartiles; whiskers, 1.5 times the interquartile range. **d** Pearson's $r$ for C–U correlation in animals based on Y20 ($x$ axis) is highly correlated with the corresponding value obtained with B20 ($y$ axis). The red line indicates where $y = x$. **e** Correlation between the biosynthetic costs of NEAAs in humans ($y$ axis) against those in yeast ($x$ axis). The nine AAs that can be synthesized from basic metabolites produced during glycolysis and TCA cycle (Ala, Asp, Asn, Arg, Gln, Glu, Gly, Pro, and Ser) are shown in red. The red line shows the results of the linear regression of biosynthetic costs of the nine AAs in humans against those in yeast. Biosynthesis of cysteine (Cys) and tyrosine (Tyr) depends on EAAs methionine and phenylalanine, respectively, and are displayed in gray. A significant correlation was still observed when incorporating Cys and Tyr in the analysis (Pearson's $r = 0.79$ and $P = 0.004$ for all 11 NEAAs). **f** C–U anticorrelation in animals is weaker using H11 metric compared with Y20 metric (Wilcoxon's signed-rank test, $P = 3 \times 10^{-61}$). The red line indicates where $y = x$

(obtained through the food chain), both of which are shaped by the H11 cost metric, presumably due to metabolic efficiency; and (2) AAs ultimately taken from autotrophs, which are constrained by the B20 or Y20 cost metric. Although it is difficult to determine the relative contribution of each source to the total AAs, our simulations (Supplementary Methods) suggest that the mixtures of AAs from the two sources always yield significantly negative correlations between the overall abundance and cost of all 20 AAs in autotrophs (Fig. 2a and Supplementary Fig. 6). Furthermore, the experimental data show that the cost is significantly anticorrelated with the abundance of free AAs in the livers of humans, chimpanzees, rhesus monkeys, and

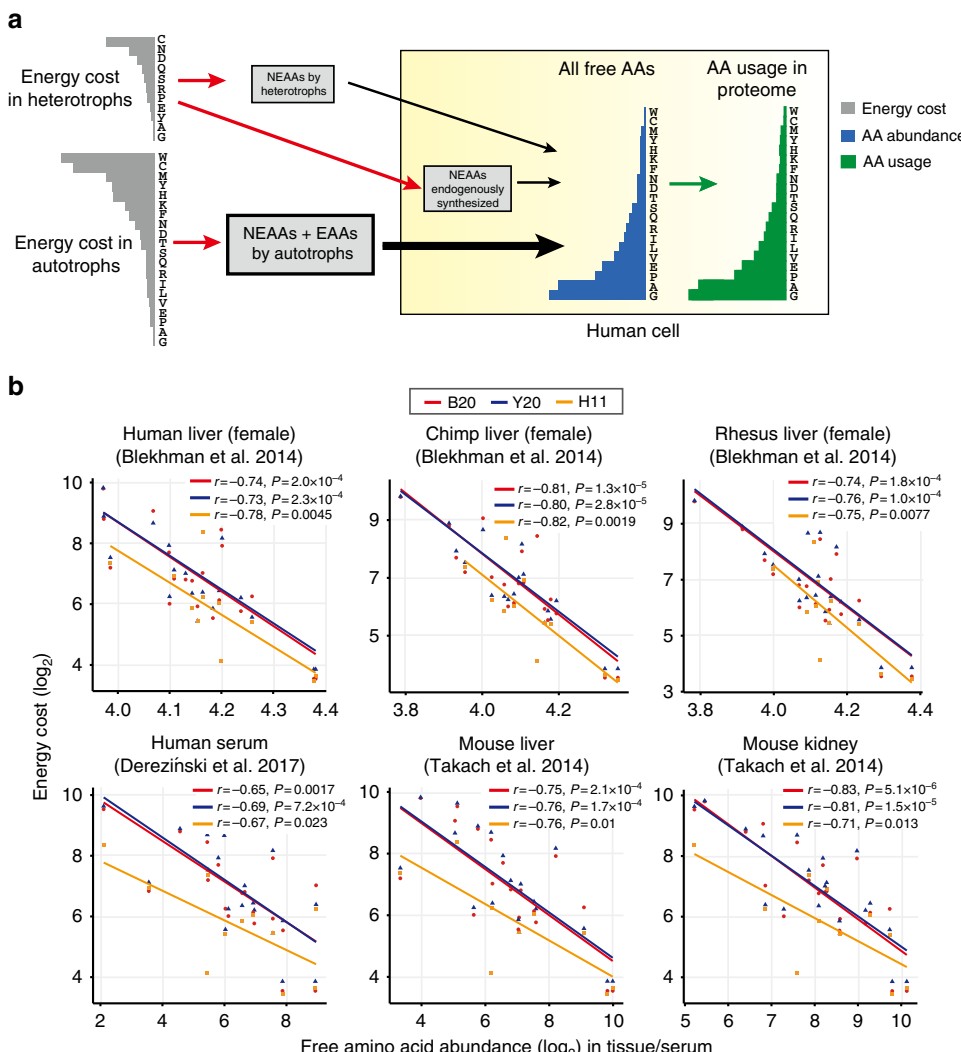

**Fig. 2** Biosynthetic cost of AAs constrains their usage in mammalian proteomes. **a** A model that explains anticorrelation between the usage of AAs in human proteomes and their cost in autotrophs (B20 or Y20) and heterotrophs (H11). Free AA pool in human cells comes from two sources: (1) NEAAs that are endogenously synthesized in human or other animal cells, which are constrained by H11 cost metric; and (2) AAs ultimately taken from autotrophs, which are constrained by B20 or Y20 cost metric. As a result, the total free AAs show anticorrelation with cost in heterotrophs (H11) or cost in autotrophs (B20 or Y20). Bioavailability of free AAs further shapes AA usage in human proteomes by optimizing compositions of protein sequences and expression levels of genes during evolution. **b** The relationship between the biosynthetic cost of AAs (B20, Y20, H11) and experimentally measured in vivo concentration of free AAs in mammalian tissues

mice[33,34], as well as in human serum[35] and mouse kidney[34] (Pearson's $r \leq -0.72$, $P < 0.05$ for B20, Y20, or H11 in each sample; Fig. 2b, Supplementary Fig. 5d). Therefore, our model suggests that the biosynthetic costs of all 20 AAs constrain the relative abundances of free AAs in human cells, which further shape AA usage in the proteomes by optimizing protein sequences and gene expression levels during evolution (Fig. 2a).

**Profound impact of AA energy costs on gene expression.** Using messenger RNA (27 tissues) or protein (30 tissues or cell types) expression data from normal human tissues, we confirmed very strong negative correlations between the biosynthetic cost and expression-normalized abundance of AAs in each tissue (Pearson's $r < -0.80$, $P < 10^{-4}$ in each test; Supplementary Fig. 7a for Y20 and Supplementary Table 3 for B20 and H11). Therefore, incorporating gene expression information further justified the impact of biosynthetic energy costs on the usage of AAs in human proteomes of different tissues. Next, we

investigated whether and how the biosynthetic cost of AAs affects human gene expression profiles by introducing the $ECPA_{gene}$ parameter (Fig. 3). For each gene, we calculated $ECPA_{gene}$ based on its protein sequence and the biosynthetic cost (B20, Y20, or H11) of individual AAs (Fig. 3a). Due to the difference in AA content, $ECPA_{gene}$ varied considerably from gene to gene (Fig. 3a), with the genes with lower $ECPA_{gene}$ significantly enriched in the pathways constitutively expressed in the cell types, and the genes with higher $ECPA_{gene}$ significantly enriched in the pathways such as gene regulation (Supplementary Table 4). Intriguingly, we detected significant negative correlations between $ECPA_{gene}$ and gene expression levels in each tissue after we grouped the expressed genes into 100 bins with increasing expression levels in that tissue (with Y20 metric, Spearman's $\rho$ ranges from $-0.766$ to $-0.345$, $P < 0.001$ in each tissue for mRNA data, and $\rho$ ranges from $-0.622$ to $-0.198$, $P < 0.05$ in the tissues, except for fetal gut and platelets for protein data, Fig. 3c and Supplementary Fig. 8; see Supplementary Table 5 for results based on B20 and H11). Of

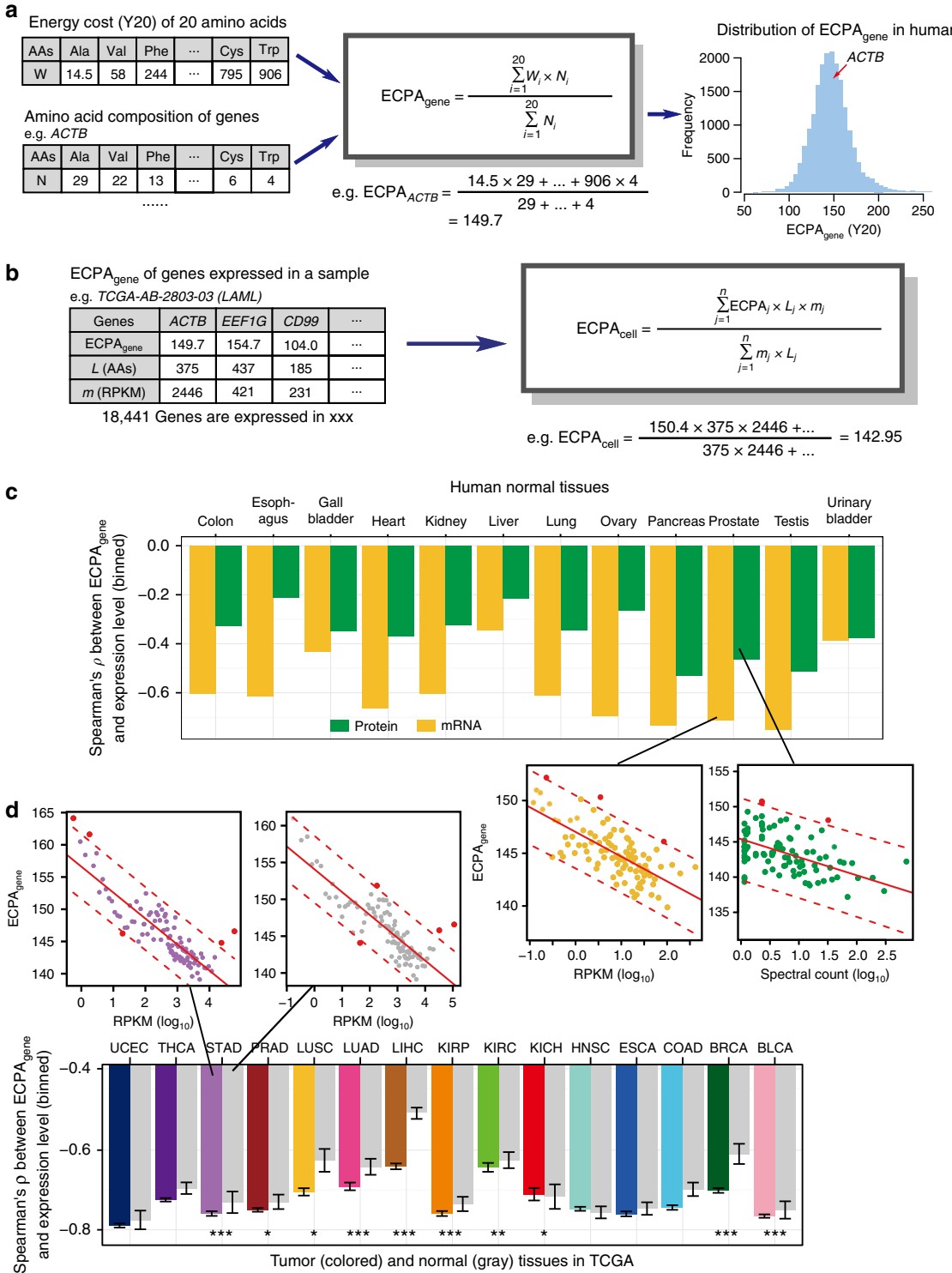

note, in the above analysis, the correlation between $ECPA_{gene}$ and protein abundance is in general weaker than that between $ECPA_{gene}$ and mRNA abundance (Fig. 3c and Supplementary Fig. 8), presumably because gene expression measured by mRNA sequencing (mRNA-Seq) is more comprehensive and accurate than the proteomic abundance quantified by mass spectrometry[36]. Overall, our results suggest that the genes highly expressed in human tissues tend to avoid the AAs that would require more energy to synthesize or which are at relatively lower abundance from exogenous supplies.

We extended this analysis to the mRNA expression data of The Cancer Genome Atlas (TCGA)[5] and confirmed similar significant negative correlations in both normal and cancer samples (only cancer types with > 10 matched normal–tumor sample pairs were analyzed; Fig. 3d and Supplementary Fig. 7b for Y20, and Supplementary Table 6 for B20 and H11 results). In 9 of the 15 cancer types surveyed, the negative correlation patterns were significantly stronger in cancer compared with normal tissues (another 4 cancer types showed similar trends, but the differences were not statistically significant; Fig. 3d). Previous results suggest

**Fig. 3** Impact of ECPA$_{gene}$ on the expression of individual genes in normal and cancer tissues. **a** Schematic diagram showing the calculation of ECPA$_{gene}$. For each gene, ECPA$_{gene}$ is the average of the biosynthetic cost of AAs weighted by the occurrence of each AA in the protein sequence. *ACTB* gene is used as an example. The histogram on the right shows the distribution of ECPA$_{gene}$ of 19,571 unique protein-coding genes in humans. **b** Illustration of ECPA$_{cell}$ calculation with mRNA-Seq data of sample TCGA-AB-2803-03 from TCGA study of acute myeloid leukemia (LAML). ECPA$_{cell}$ is an average of ECPA$_{gene}$ of all expressed genes weighted by lengths regarding encoded AAs and expression levels of those genes. **c** Correlations between ECPA$_{gene}$ and gene expression level in 12 normal human tissues with both mRNA-Seq and proteomic data available. For each tissue, genes were divided into 100 groups based on their expression levels (spectral count for proteomic data and RPKM for mRNA-Seq), and the median expression level ($\log_{10}$) and median ECPA$_{gene}$ in each group were used in the correlation analysis. Two representative correlations are magnified for more detail. **d** Correlations between ECPA$_{gene}$ and gene expression level across different cancer (colored) and normal tissues (gray) using TCGA mRNA-Seq data. For each sample of each cancer type, genes were divided into 100 groups based on their expression levels and, the median expression level and median ECPA in each group were used in the correlation analysis. Error bars indicate the 95% confidence intervals of $\rho$. The number of tumor and normal tissue samples for each cancer type can be found in Supplementary Table 6. For each cancer type, the significant difference in the correlation coefficient (Spearman's $\rho$) between tumor and related normal samples is marked as *$P < 0.05$; **$P < 0.01$; and ***$P < 0.001$. Two representative correlations for tumor and normal samples of STAD are magnified for more detail

cancer patients usually have dysregulated AA levels in blood[37–39] or tumor tissues[40,41]. However, we observed similar negative correlations between the cost and abundance of the free AAs in tumor and matched normal tissues for a variety of cancer types (Supplementary Fig. 9 and Supplementary Table 7). These results suggest that cancer cells may more efficiently manage protein synthesis using the AAs available within their microenvironment.

**Consistent prognostic power of ECPA$_{cell}$ across cancer types.** We next questioned whether cancer cells utilize AA for protein synthesis in a way that is more economical than that of normal cells. We analyzed the relationship between ECPA$_{gene}$ and the fold-change in protein abundance in invasive breast carcinoma relative to matched normal samples that were measured with quantitative mass spectrometry in a previous study[42]. After grouping proteins into equal-sized bins based on increasing difference, we found that the change in protein abundance in the tumor relative to normal cells is inversely correlated with ECPA$_{gene}$ for tumors with (Spearman's $\rho = -0.42$, $P = 0.0023$) or without (Spearman's $\rho = -0.32$, $P = 0.022$) lymph node metastasis (Supplementary Fig. 10). These results support the hypothesis that cancer cells utilize AAs for protein synthesis more economically by (1) preferentially downregulating or (2) avoiding upregulating the genes rich in biosynthetically expensive AAs, or by both mechanisms.

To more generally study the impact of managing AA usage in various cancer types, we performed a pan-cancer analysis based on ECPA$_{cell}$ (Fig. 3b) using TCGA mRNA-Seq expression data of 33 cancer types (Supplementary Fig. 11). Of note, ECPA$_{cell}$, which measures the virtual average cost of proteinogenic AAs in the cells, not only considers the composition of AAs in the protein sequences but also incorporates the gene expression levels. As TCGA mRNA-Seq expression data were quantified at the tissue level, the ECPA$_{cell}$ value represents the average virtual cost of proteinogenic AAs across all the cells present in that sample. We obtained very similar results with the B20, Y20, or H11 cost metric in the analyses. In the following, we primarily focused on the results based on Y20, as it included all 20 AAs. In 11 of the 15 cancer types that have mRNA expression data available for at least 10 normal samples, ECPA$_{cell}$ was significantly lower in tumors than in normal tissues (Fig. 4a), suggesting that reducing the usage of more expensive AAs in protein synthesis is a general trend for cancer cells. Within a cancer type, the gene expression profiles of different patients are highly heterogeneous. Hence, we analyzed the ECPA$_{cell}$ of tumor samples from different subtypes of breast carcinoma[43], which has the largest number of samples in TCGA data. Compared with the normal samples, all tumor subtypes have significantly lower ECPA$_{cell}$ (Supplementary Fig. 12), suggesting that the reduced ECPA$_{cell}$ in cancer cells is

robust with respect to tumor subtype. To assess the influence of the heterogeneous cellular composition in the cancer samples[6], we performed ECPA$_{cell}$ analysis on previously published single-cell RNA sequencing (RNA-Seq) data of melanoma[44] and ovarian carcinoma cells[45]. For both cancer types, the cancer cells have significantly lower ECPA$_{cell}$ values than the immune or stromal cells (Supplementary Fig. 13), suggesting that the reduced ECPA$_{cell}$ in tumors is mainly influenced by the malignant cells rather than the immune and stromal cells within the tumor microenvironment. As the number of AA changes caused by somatic mutations in a cancer sample is small (20~100)[5], such AA changes have negligible effects on the observed difference in ECPA$_{cell}$ values between the normal and cancer samples. Indeed, we validated this hypothesis by considering the somatic mutations and calculating the ECPA$_{cell}$ values in each tumor sample (Supplementary Fig. 14).

To test whether the cancer samples with reduced usage of expensive AAs (i.e., lower ECPA$_{cell}$) are more aggressive, we compared the ECPA$_{cell}$ of tumor samples from patients diagnosed at different pathologic stages (from I to IV, see Methods). We found negative correlations between ECPA$_{cell}$ and tumor stage in 16 of the 19 cancer types that have pathological stage information available, 9 of which were statistically significant (Fig. 4b). We further confirmed significant negative correlations between ECPA$_{cell}$ and pathologic stages in the 9 cancer types (empirical $P < 0.05$ for each cancer type, Supplementary Fig. 15) with permutation tests by shuffling ECPA$_{cell}$ among samples 10,000 times and repeating the correlation analysis (Methods). Therefore, utilizing AAs more economically in protein synthesis confers a greater proliferation advantage upon cancer cells.

Next, we considered whether ECPA$_{cell}$ is associated with patient survival time. Focusing on 17 cancer types with sufficient samples and events (Methods and Supplementary Fig. 11), we found that patients with lower ECPA$_{cell}$ showed significantly worse survival probability compared with those with higher ECPA$_{cell}$ in nine cancer types and we did not find a significantly reversed pattern in any cancer type (split by the median ECPA value, log-rank test, Fig. 4c, d). Further, a lower ECPA$_{cell}$ was significantly associated with poor survival using a univariate Cox proportional hazards model in the nine cancer types. Collectively, in 11 of the 17 cancer types surveyed, lower ECPA$_{cell}$ showed a significant correlation with poorer patient prognosis by either log-rank test or Cox model (see additional cancer types in Supplementary Fig. 11). To confirm the statistical significance of the observed pattern, we performed permutation tests on cancer samples and found that the number of cancer types with consistent survival correlation was much higher than the random expectation (at most five in permutations, $P < 2 \times 10^{-4}$, Supplementary Fig. 16a). Importantly, in six cancer types, the

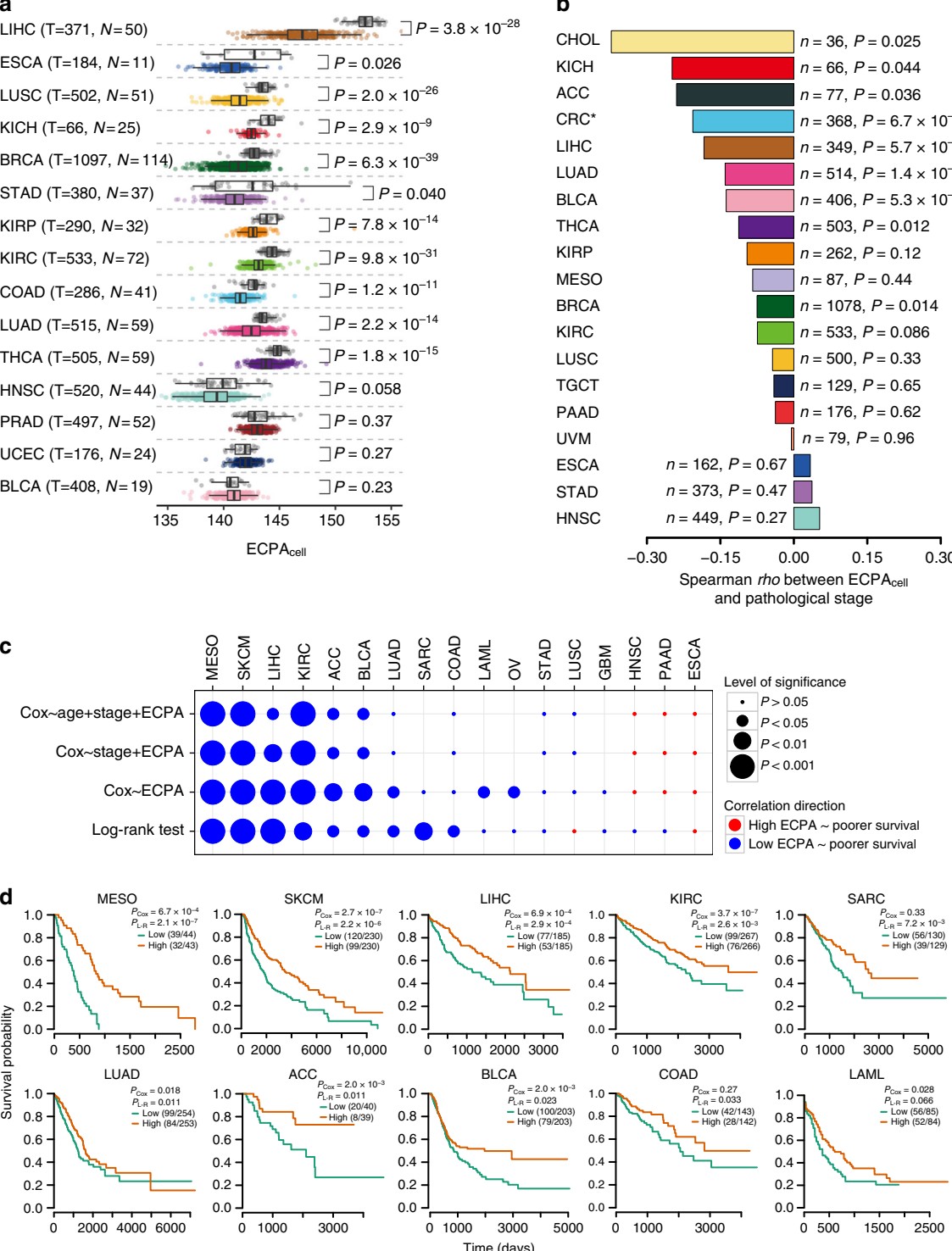

**Fig. 4** Clinically relevant patterns of ECPA$_{cell}$ across cancer types. **a** Boxplot showing ECPA$_{cell}$ of tumor samples and matched normal tissue samples in 15 cancer types for which mRNA-Seq data of > 10 normal samples were available. The number of tumor samples (T), the number of normal samples (N), and Wilcoxon's rank-sum test *P*-values are displayed in the plot. Center line, median; box limits, upper and lower quartiles; whiskers, 1.5 times the interquartile range. **b** Bar plot showing Spearman's correlation coefficient of ECPA$_{cell}$ and the pathologic stage for patients with 19 cancer types. The numbers of tumor samples (*n*) and Spearman's rank correlation *P*-values are displayed in the plot. *Colon and rectal adenocarcinoma are merged as colorectal carcinoma (CRC) in the analysis. **c** Associations between ECPA$_{cell}$ and the patients' survival times using either log-rank tests or Cox proportional hazards model in 17 cancer types that have ≥ 75 samples and ≥ 25% events. Sample size and results for additional cancer types are provided in Supplementary Fig. 11. Circle size indicates the significance of the correlation; color indicates correlation direction. **d** Kaplan–Meier plots showing the survival probability of patients with lower ECPA$_{cell}$ or higher ECPA$_{cell}$ in ten cancer types. For each cancer type, patients were divided into two equal groups based on ECPA$_{cell}$ of the patients' tumor samples. *P*-values of log-rank and univariate Cox tests are shown

association with ECPA$_{cell}$ remained significant even when the pathologic tumor stage and patient age were considered in the multivariate analysis (Fig. 4c), which suggests that ECPA$_{cell}$ provides additional prognostic power over clinical variables. For comparison, we stratified patients by the expression level of individual genes and tested their associations with the pathological stage or patient survival time. Among 18,919 genes surveyed, only one gene (*LOX*) showed a comparable, consistent association with both pathological stage and survival analysis, and the probability of a gene with similar prognostic power across multiple cancers was $2.1 \times 10^{-4}$ (Supplementary Fig. 16b). Indeed, when examining a set of known cancer therapeutic targets or biomarker genes[46], none of them showed such a consistent prognostic pattern as ECPA$_{cell}$ (Supplementary Fig. 16c). Notably, we also repeated the whole pan-cancer analytical procedures with B20 or H11 and obtained overall patterns that were very similar to those for Y20 (Supplementary Figures 17–20). As tumors often experience hypoxia[47] and thus obtain part of their cellular energy via fermentation[48], we also repeated the pan-cancer association analysis with anaerobic costs of AAs and found that our conclusions still held (Supplementary Figures 21–23). Overall, our results indicate that tumors with lower ECPA$_{cell}$ tend to be more aggressive, and patients with such tumors have shorter survival times across a broad range of cancer types. These results also highlight the feasibility of ECPA$_{cell}$ as a potential prognostic marker for patient stratification.

**Reduced ECPA in experimental evolution of xenograft tumors.** Based on our observations, we argue that lower ECPA$_{cell}$ may be an important feature shaped by natural selection at the systemic and cellular level, and the trend will be enhanced during the evolution of a tumor. To test this hypothesis, we analyzed the data generated in an experimental evolution of xenograft tumors in which an early transformed cell population was first obtained by introducing a mutated oncogene, HRAS$^{V12}$, into a normal human breast epithelial cell line (MCF10A)[49]. These MCF10A-HRAS cells were xenografted into mice to form the first-stage xenograft tumor (XT1), the subsequent second-stage xenograft tumor (XT2), then XT3…, until the metastatic tumor was detected in the mouse carrying XT8. The sequential cell samples collected from MCF10A-HRAS, XT1 to XT8, and the two metastatic tumors, XT8_M1 and XT8_M2, represent the full evolutionary process from tumor initiation to metastasis. We analyzed the mRNA-Seq data of the nine primary tumors (Methods) and found that ECPA$_{cell}$ is reduced in the xenograft tumors (XT1 to XT8) compared with the ancestral MCF10A-HRAS cells. Strikingly, we observed a clear decreasing trend of ECPA$_{cell}$ in a temporal order of the eight xenograft tumors (XT1 to XT8) (Pearson's $r \le -0.81$, $P < 0.05$ for each cost metric; Fig. 5a). This in vivo experimental study supports that ECPA$_{cell}$ is selected for reduction during tumor evolution.

To examine the key factors affecting the evolutionary process for managing AA usage, we conducted simulations on the evolution of the ECPA of a single cancer cell population, in which the ECPA$_{cell}$ of the cells varied at a rate of $\nu$ (per generation). We then sampled them to the next generation based on their fitness values given a selective strength of $s$ (Methods, Fig. 5b and Supplementary Fig. 24). We found that higher $\nu$ and stronger $s$ can lead to a quicker decrease in ECPA$_{cell}$, whereas the speed of the decrease is largely determined by $s$ (Fig. 5c, d). We note that the reduction in ECPA$_{cell}$ is not necessarily linearly correlated with the selective advantages in this simulation. Collectively, both our experimental evolution and simulations suggest that reduced ECPA$_{cell}$ is an important feature of tumor cells during cancer progression.

**Biological themes related to reduced ECPA$_{cell}$ in tumors.** To test whether the reduced ECPA$_{cell}$ in cancer cells occurs by expression level changes of genes of certain pathways or at the genome-wide level, we systematically searched for genes that had expression levels correlated with ECPA$_{cell}$ among the samples for 31 TCGA cancer types that have at least 50 samples available (Methods). As expected, for each cancer type, the positively correlated genes overall have higher ECPA$_{gene}$, and the negatively correlated genes tend to have lower ECPA$_{gene}$ (Fig. 6a, Supplementary Fig. 25, Supplementary Tables 8 and 9). For most cancer types, the positively correlated genes are significantly enriched in the pathways related to the mitochondrion (Fig. 6b, Supplementary Table 10). The negatively correlated genes are over-represented in pathways that tend to have lower ECPA$_{gene}$ compared to the genomic background (Fig. 6c, Supplementary Table 10), and the power of ECPA$_{cell}$ in the pan-cancer analysis was considerably compromised when we excluded these pathways (Supplementary Fig. 26 and Supplementary Table 11). These results suggest that the pathways rich in expensive AAs are not upregulated overall in cancer cells so that expensive AAs are economically used. However, we did not find such patterns for the pathways enriched with positively correlated genes (Supplementary Table 11), suggesting that reduced ECPA$_{cell}$ in tumor cells is not the direct consequence of the downregulation of genes in specific pathways.

Tumor suppressors and cancer drivers[4], as well as genes involved in AA biosynthesis and transport[50,51], are often dysregulated in tumor cells. Accordingly, we identified numerous genes in those functional categories that are differentially expressed in tumor cells (Fig. 6d and Supplementary Fig. 27a–d). Nevertheless, the dysregulation of these genes is unlikely to predominantly affect ECPA$_{cell}$ in tumors, as they have ECPA$_{gene}$ that is similar to the background level (Supplementary Fig. 27e); and importantly, the results of the overall pan-cancer analysis remain intact after we excluded each category from the analysis (Supplementary Table 11). The expression levels of proliferation-related genes[52] are increased in tumors compared to the matched normal samples (Supplementary Fig. 28a). Although the proliferation-related genes have lower ECPA$_{gene}$ than the genomic background (Supplementary Fig. 28b), the results of pan-cancer analyses are only slightly affected by these genes (Supplementary Fig. 29 and 30). Furthermore, the reduction of ECPA$_{cell}$ during experimental evolution of xenograft tumors still holds when the proliferation-related genes were excluded (Supplementary Fig. 31). These results suggest that the association between ECPA$_{cell}$ and cancer progression is unlikely to be caused by changes in proliferation-related genes alone. To test whether cancer cells preferably express proteins with lower total biosynthetic cost, we calculated the total energy cost of each protein (EC$_{gene}$) as the sum of the biosynthetic cost of AAs in each protein sequence. As expected, genes with higher EC$_{gene}$ tend to have lower expression levels in both normal tissues and tumors (Supplementary Fig. 32a), and be under-represented in the upregulated genes in cancer cells (Supplementary Fig. 32b). Moreover, the EC$_{cell}$ values, which are calculated as the average EC$_{gene}$ of genes weighted by their expression levels (Methods), are significantly lower in tumors than in normal tissues (Supplementary Fig. 33). Nevertheless, the pathological stage of tumors or the survival time of patients is generally not associated with the EC$_{cell}$ parameters in the pan-cancer analysis (Supplementary Fig. 33), suggesting that EC$_{cell}$ is not suitable for a prognostic marker of cancer progression. Taken together, our results suggest that the economical use of AAs in protein synthesis in cancer cells is achieved by (1) avoiding upregulation of pathways enriched for expensive AAs and (2) the cumulative effect of downregulating individual genes that are enriched for expensive AAs. We

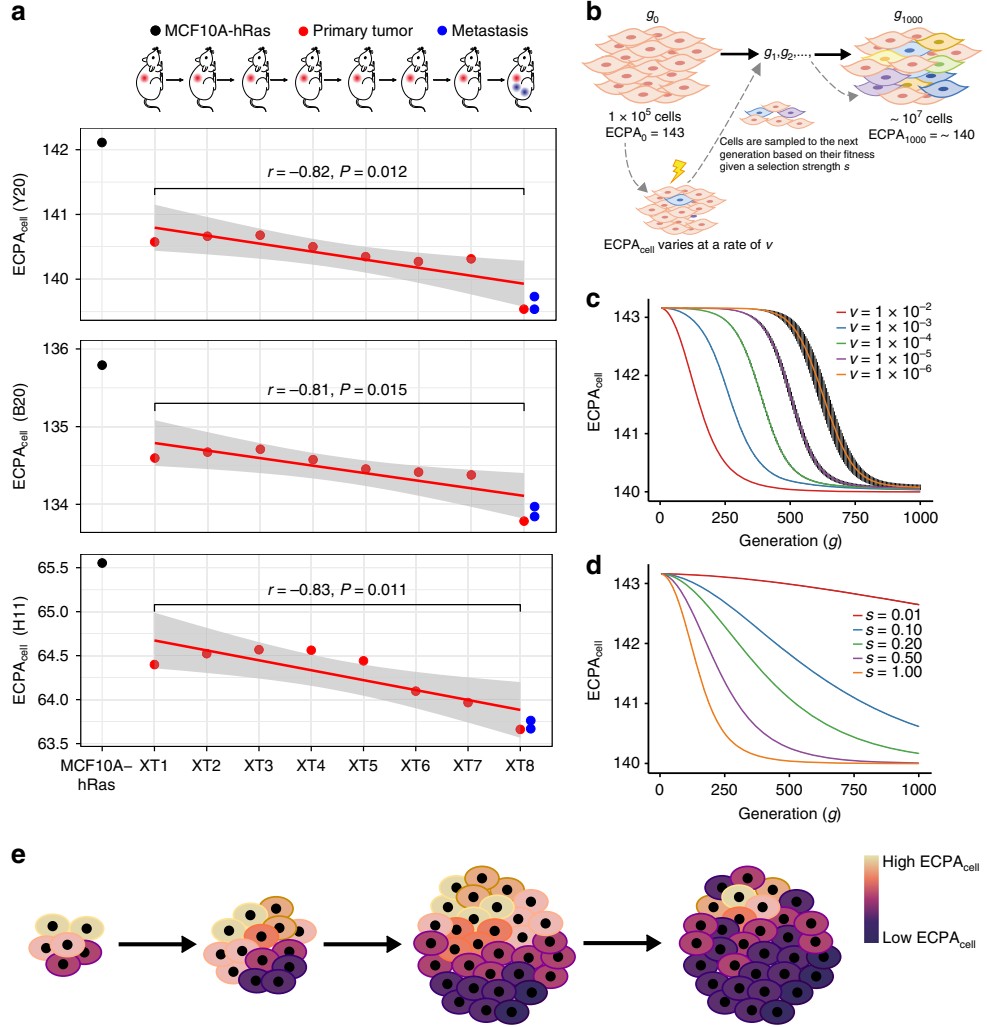

**Fig. 5** ECPA$_{cell}$ change during the evolution of a single cancer cell population. **a** The decreasing trend of ECPA$_{cell}$ during the experimental evolution of a xenograft tumor. The MCF10A-HRAS cells (in black) that were xenografted into mice for generations. XT1, XT2, ..., XT8 represent the first-stage xenograft tumor, the second-stage, ..., the eighth-stage (in red); two metastatic tumors were detected in the mouse carrying XT8 (in blue). P-values for linear regression of ECPA$_{cell}$ against generation number (XT1 to XT8) are shown. **b** Computational simulation setup for the evolutionary process of a single tumor cell population based on the selection of ECPA$_{cell}$ value of each cell in the population. **c** Mean ECPA$_{cell}$ trend of a single cancer cell population under different mutation rates $v$ that with fixed selective strength ($s = 1$) throughout the simulation. **d** Mean ECPA$_{cell}$ trend of a single cancer cell population under different selective strengths $s$ with a fixed mutation rate $v = 1 \times 10^{-6}$ throughout the simulation. **e** Cartoon showing that ECPA$_{cell}$ of a cancer cell population gradually decreases under selection for increased AA metabolic efficiency

conclude that the efficient use of AAs in cancer cells is achieved by the coordinated regulation of gene expression at the whole-transcriptome level. Although specific pathways might contribute to this process, none of them is overwhelmingly dominant in this process.

**The predictive power of ECPA$_{cell}$ for immunotherapy.** Checkpoint inhibitor immunotherapy is one of the most exciting developments in cancer treatment[53]. The expression levels of *PD-1* (*PDCD1*) or *PD-L1* (*CD274*) are associated with the response to checkpoint blockade therapy[54,55]. Although *PD-1* and *PD-L1* are usually dysregulated in tumors compared to normal tissue samples (Supplementary Figs. 34a and 35a), the expression level of neither gene showed consistent association with the pathological stage of tumors or patient survival time (Supplementary Fig. 34 and 35). We questioned whether ECPA$_{cell}$ can predict response to immunotherapy and hypothesized explicitly that higher ECPA$_{cell}$ is associated with a better clinical outcome. We applied our method to a recent study on anti-PD-1 therapy in metastatic

melanoma[56] in which the mRNA-seq data for patient samples are available. Indeed, ECPA$_{cell}$ for patients in the responding group was significantly higher than that of patients in the non-responding group (one-sided *t*-test, $P = 0.032$, Fig. 7a). By contrast, we did not find significant differences in the expression levels of *PD-1* (*t*-test, $P = 0.85$) or *PD-L1* (*t*-test, $P = 0.49$) between patients in the responding group and the non-responding group, which is consistent with a recent study[57]. These results suggest that tumors with low ECPA$_{cell}$ can survive better than those with high ECPA$_{cell}$ when undergoing a T-cell attack and therefore become more resistant to immunotherapies. To further confirm that the observed significant pattern is due to the biosynthetic costs of different AAs, we randomly permutated the biosynthetic energy costs of AAs 1000 times, repeated the above analysis between the two response groups, and visualized the obtained P-values and ECPA$_{cell}$ differences [log$_2$(responding/non-responding)] using a volcano plot (Fig. 7b). We found that the ECPA$_{cell}$ difference obtained from using the real biosynthetic energy costs of AAs was significantly larger than that obtained

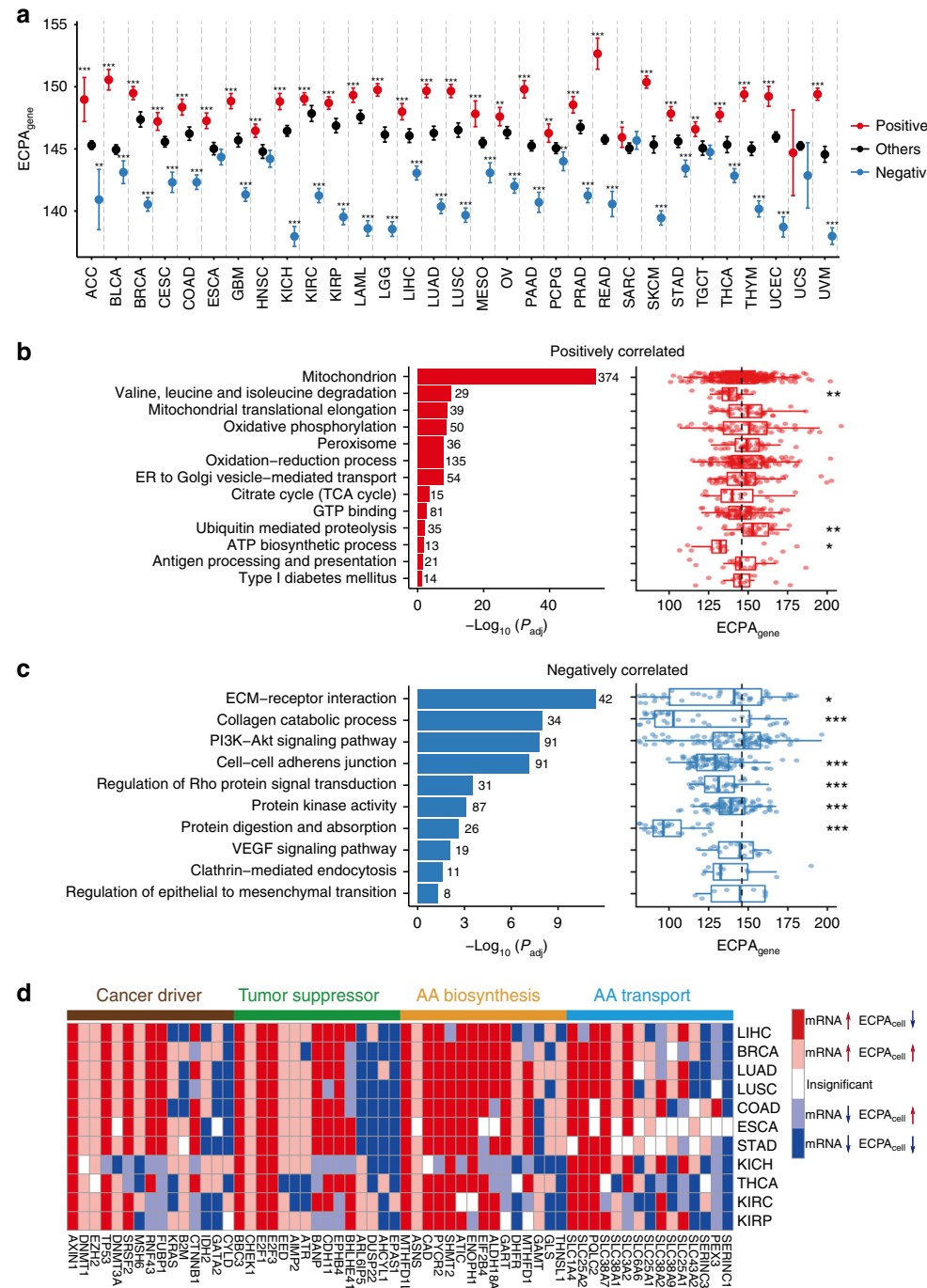

**Fig. 6** Genes and pathways associated with ECPA$_{cell}$ across 31 TCGA cancer types. **a** Distribution of ECPA$_{gene}$ of the genes that had expression levels positively (red) or negatively (blue) correlated with ECPA$_{cell}$ among samples (FDR-adjusted $P < 0.05$) and the other genes (black) in each of the 31 cancer types with at least 50 samples. The number of positively or negatively correlated genes is presented in Supplementary Table 8. Error bars indicate 95% confidence intervals. Wilcoxon's rank-sum tests were performed to compare the ECPA$_{gene}$ of positively or negatively correlated genes and that of the remaining genes (*$P < 0.05$; **$P < 0.01$; ***$P < 0.001$). **b** Pathways over-represented in positively correlated genes and the distribution of ECPA$_{gene}$ of genes in each pathway (number of genes displayed beside the bar). ECPA$_{gene}$ of positively correlated genes in each pathway compared to genomic background (dashed line) with Wilcoxon rank-sum tests. **c** Pathways over-represented in negatively correlated genes and the distribution of ECPA$_{gene}$ of genes in each pathway (number of genes displayed beside the bar). ECPA$_{gene}$ of negatively correlated genes in each pathway compared to genomic background (dashed line) with Wilcoxon's rank-sum tests. **d** Examples showing differential expression of cancer drivers, tumor suppressors and genes related to AA biosynthesis or transport between tumor and normal samples with respect to their ECPA$_{gene}$ in the 11 cancer types that had significantly lower ECPA$_{cell}$ in tumors. Up- or downregulated genes are identified with $t$-tests at an FDR of 0.05 and displayed in red and blue, respectively. Differential expression events that contribute to the decrease or increase of ECPA$_{cell}$ in tumors are displayed with dark and light color, respectively. Insignificant events are shown in white. For box plots, center line, median; box limits, upper and lower quartiles; whiskers, 1.5 times the interquartile range

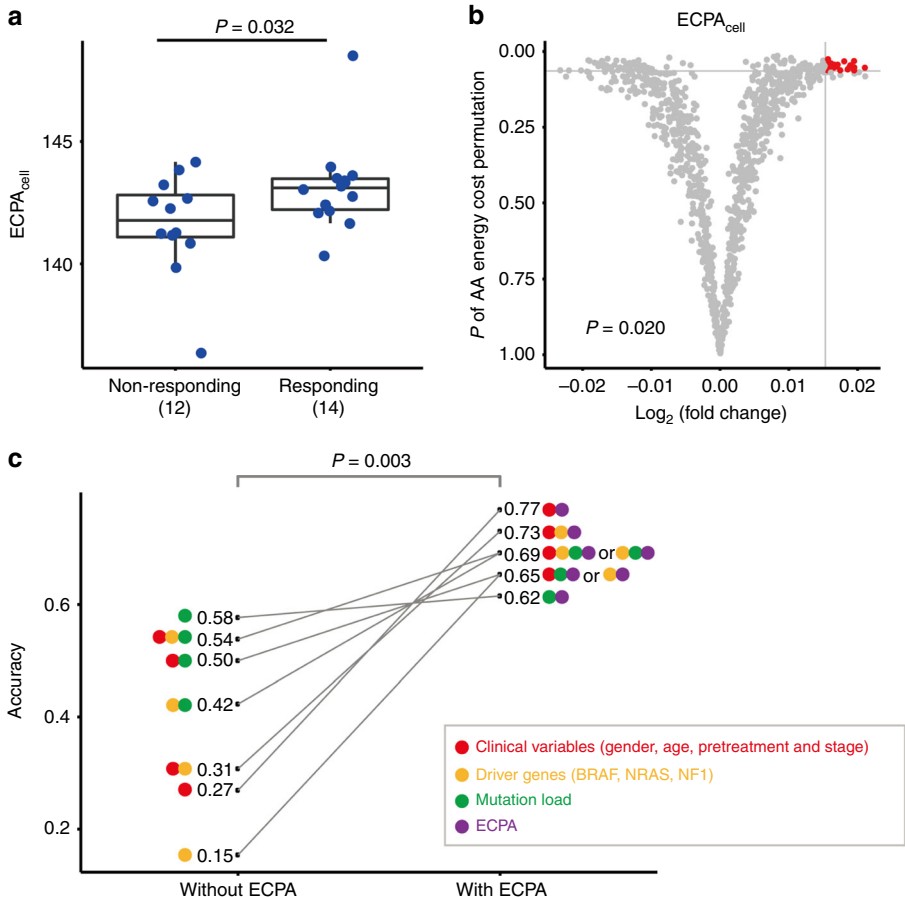

**Fig. 7** The predictive power of ECPA_cell for response to anti-PD-1 immunotherapy. **a** Comparison of ECPA_cell between responding (14 patients) and non-responding (12 patients) groups diagnosed with melanoma. One-sided *t*-test *P*-value is shown. Center line, median; box limits, upper and lower quartiles; whiskers, 1.5 times the interquartile range. **b** Volcano plots showing how *P*-values and ECPA_cell differences (responding/non-responding) for the two-group comparison of ECPA_cell are distributed given 1000 permutations, where the biosynthetic energy costs of 20 AAs were randomly shuffled. The gray horizontal and vertical lines indicate the *P*-value and the fold-change observed from the true ECPA_cell. The red dots falling in the upper-right corner of the gray lines represent random cases that are better than the true values shown in **a**. Empirical *P*-value (*P* = 0.02) was estimated using the number of red dots divided by the total number of permutation tests. **c** Comparison of predictive power between the models with and without ECPA_cell using random forests with leave-one-out cross-validation. In addition to ECPA_cell (purple circle), three groups of candidate features were used: clinical variables (red circle), mutation status of melanoma driver genes (yellow circle) and mutation load (green circle). The *P*-value (0.003) was calculated by paired *t*-test between the models with and without ECPA_cell as the candidate feature. The paired models are linked by the solid gray lines

from using the shuffled energy costs (empirical *P* = 0.012, Fig. 7b).

We further examined whether ECPA_cell can improve the predictive power of clinical variables in response to immunotherapy using the common machine learning method of random forests[58] with leave-one-out cross-validation (Fig. 7c). We split the candidate features into four groups: (i) clinical variables (i.e., gender, age, pretreatment, and pathologic stage); (ii) mutation status of three well-known melanoma driver genes (*BRAF*, *NRAS*, and *NF1*); (iii) mutation load; and (iv) ECPA_cell. Without ECPA_cell, the mutation load alone achieved the best accuracy (0.58) among all the models. After adding ECPA as candidate features into the models, there was significantly improved predictive power across the models (median accuracy 0.69 [with ECPA_cell] vs. 0.42 [without ECPA_cell]; one-sided paired *t*-test, *P* = 0.003). The best predictive model was the combination of clinical variables and ECPA_cell, with a predictive accuracy of 0.77. These results show that tumors with high ECPA_cell are more responsive to anti-PD-1 immunotherapy, and this feature can significantly improve the predictive power of any combination of clinical data, signature genes, and mutation load. Thus, ECPA_cell represents a novel,

simple, and promising metric for predicting the response to checkpoint inhibitor immunotherapy.

## Discussion

Cancer cells employ multiple strategies to acquire AAs[10], such as the endogenous synthesis of NEAAs[11,19,20,59,60], upregulation of AA transport[50,51,59], or through micropinocytosis[61]. Besides protein synthesis, certain AAs, such as asparagine[19,59,60], glycine[20], glutamine[11,18,62], histidine[63], leucine[64], proline[65], and serine[66], participate in various cellular processes such as nucleotide synthesis, cellular signaling, and regulation of gene expression[67,68]. Of note, recent studies have demonstrated that protein synthesis is the cellular process that consumes the most AAs[11]. As the use of all 20 AAs in human proteomes is constrained by their synthetic costs in living organisms, our ECPA concept effectively reflects how cancer cells optimize gene expression profiles for AA usage adaptation. We revealed a common principle governing cancer evolution: cancer cells evolve to use AAs more economically by downregulating genes that are rich in costly AAs. This trend is evident through the comparison between tumor and normal tissue samples, the within-disease

analysis across a diversity of cancer types, and the in vivo experimental evolution of a xenograft tumor. Thus, our study provides novel insights into how efficient usage of AAs benefits cancer cells from an evolutionary perspective and at the systemic level. Moreover, the ECPA$_{cell}$ metric we developed shows good prognostic power (compared to individual genes) across many cancer types and can also help predict the tumor response to anti-PD-1 therapy for patients with metastatic melanoma.

As the ECPA$_{cell}$ metric is designed to quantify the usage of AAs in protein synthesis based on their biosynthetic costs, by definition the most appropriate approach for calculating ECPA$_{cell}$ should be the rate of protein synthesis. In this study, we calculated ECPA$_{cell}$ using RNA-Seq data, as recent studies based on ribosome profiling have demonstrated a high correlation between mRNA and the rate of protein synthesis[69]. To further validate our analysis, we retrieved the ribosome profiling data of normal ($n =$ 6) and tumor ($n = 10$) samples of human kidney tissue[65] and calculated ECPA$_{cell}$ with the ribosome-protected fragments (Supplementary Methods). Consistent with the observation using TCGA mRNA-Seq data, we found that ECPA$_{cell}$ in the tumor samples is significantly lower than that in the normal samples whenever we used the Y20, B20 or H11 metric (Supplementary Fig. 36). Hence, our ECPA analysis based on mRNA-Seq data provides a simple and powerful method that informs how economically AAs are utilized during cancer evolution.

Conceptually, our ECPA study is fundamentally novel to the field and represents a substantive departure from the status quo, namely, gene-based analyses. We emphasize the management of the overall AA expenditures by summarizing the effects of all the genes in the cell, because individual changes that accumulate at the systemic level collectively define the cellular properties that are evident through natural selection in tumor evolution. From this point of view, our study emphasizes the importance of holism in understanding cancer evolution and improving cancer medicine.

## Methods

**Biosynthetic energy costs of AAs.** The biosynthetic cost of each AA, $C_i$ ($i = 1$ to 20), measured by the number of high-energy phosphate bonds required for synthesis, was obtained from previous studies in bacteria[22] and yeast[21,26]. The detailed procedures for calculating the biosynthetic cost for each of the 11 NEAAs in humans (H11) are presented in Supplementary Figures 1–3 and the Supplementary Methods. For each biosynthetic cost metric (B20, Y20, or H11), the decay rate-normalized biosynthetic cost of an AA, $W_i$ ($i = 1$ to 20 for B20 and Y20, and $i = 1$ to 11 for H11), was calculated as the product of the biosynthetic cost and the decay rate for each AA, i.e., $W_i = C_i \cdot D_i$ ($i = 1$ to 20 for B20 and Y20, and $i = 1$ to 11 for H11), as described previously[23].

The anaerobic biosynthetic cost of AAs in yeast was obtained from previous studies[21,26]. The anaerobic biosynthetic cost of AAs in bacteria or humans was calculated by counting only the number of high-energy phosphate bonds that are directly consumed or produced during AA biosynthesis as performed previously[26]. The decay rate-normalized anaerobic cost of AAs in yeast, bacteria or humans was calculated as described above.

**The C–U correlation analysis based on protein sequences.** All the protein sequences in seven taxonomic divisions (archaea, bacteria, protists, plants, fungi, invertebrates, and vertebrates) annotated in the Swiss-Prot and TrEMBL databases were downloaded from the UniProt website (www.uniprot.org). Species with more than 500 unique protein sequences were analyzed. In each species, Pearson's $r$ between the occurrence of AAs ($\log_2$) and the cost of AAs (B20, Y20, or H11) was calculated. We performed permutation tests by randomly shuffling the cost of AAs (B20, Y20, or H11) 10,000 times and repeating the correlation analysis in *Escherichia coli*, *Saccharomyces cerevisiae*, *Arabidopsis thaliana*, *Drosophila melanogaster*, *Mus musculus*, and humans.

**The relationship between in vivo concentrations and biosynthetic costs of AAs.** The in vivo concentrations of AAs hydrolyzed from proteins of bacteria, yeast, and whole bodies of different animals (Supplementary Fig. 5), as well as the in vivo concentrations of free AAs in tissues/blood of humans and other mammals (Fig. 2a), were extracted from previous studies and are summarized in Supplementary Table 12. For each sample, Pearson's $r$ between the concentrations ($\log_2$)

and cost ($\log_2$) of AAs (B20, Y20, or H11) was calculated. We also performed permutation tests by randomly shuffling the costs of AAs (B20, Y20, or H11) 10,000 times and repeating the correlation analysis in each sample.

**Calculating the energy cost per AA.** The ECPA for a gene, ECPA$_{gene}$, was calculated with the formula $\text{ECPA}_{gene} = \sum_{i=1}^{k} W_i \cdot N_i / \sum_{i=1}^{k} N_i$, where $N_i$ is the number of the AA $i$ in the protein sequence of that gene ($k = 20$ for B20 and Y20, and $k = 11$ for H11). The total energy cost of AAs in a protein sequence was thus calculated as $\text{EC}_{gene} = \sum_{i=1}^{k} W_i \cdot N_i$. The ECPA for a sample, ECPA$_{cell}$, was calculated with the formula $\text{ECPA}_{cell} = \sum_{j=1}^{n} \left( \text{ECPA}_j \cdot L_j \cdot m_j \right) / \sum_{j=1}^{n} \left( m_j \cdot L_j \right)$, where $L_j$ is the total number of AAs in the protein sequence of gene $j$, $n$ is the number of genes expressed in each sample, $m_j$ is the abundance of gene $j$ in the sample, and ECPA$_j$ is ECPA$_{gene}$ for gene $j$ with B20, Y20, or H11. Similarly, the average EC$_{gene}$ for a sample, EC$_{cell}$, was calculated as $\text{EC}_{cell} = \sum_{j=1}^{n} \left( \text{EC}_j \cdot m_j \right) / \sum_{j=1}^{n} m_j$. To control for the influence of DNA mutations in tumors, we obtained somatic mutation data of tumor samples from TCGA data portal (tcga-data.nci.nih.gov). For each tumor sample, the peptide sequence of each mutation-containing gene was corrected based on somatic mutations before calculating $N_i$, the number of the AA $i$ in the protein sequence of a gene. Then the same formula presented above was used to calculate ECPA$_{cell}$ for the sample.

**Correlation between ECPA$_{gene}$ and gene expression levels.** The quantification of mRNA expression in 27 human tissues (Supplementary Table 3) was obtained from Fagerberg et al.[70]. Protein abundances (spectral counts) of 30 human tissues and cells (Supplementary Table 5) were taken from Kim et al.[71]. The level-3 gene expression quantification in different cancer types (i.e., rsem.genes.normalized_results, except RPKM for acute myeloid leukemia [LAML] and stomach adenocarcinoma [STAD]) was downloaded from TCGA data portal (tcga-data.nci.nih.gov). For each gene, the principal splice isoform annotated by APPRIS (appris.bioinfo.cnio.es, 2016_06.v17) was employed. The proteomic data for breast cancer were taken from Pozniak et al.[42]. In the mRNA-Seq analysis, the RefSeq coding sequences (www.ncbi.nlm.nih.gov/refseq/, 2016-07-28) were translated into proteins, and the relative abundance of a protein was assumed in scale to its mRNA. For each sample, the expressed genes were divided into 100 groups based on increased expression levels, and Spearman's rank correlation coefficient $\rho$ between the median expression level ($\log_{10}$) and median ECPA$_{gene}$ in each group was calculated. For TCGA mRNA-Seq data, in each cancer type, we compared the $\rho$-values in the tumor samples versus those in the normal tissue samples with the Wilcoxon's rank-sum test.

**Analysis of clinical relevance of ECPA in TCGA datasets.** We compared the ECPA$_{cell}$ difference between tumor and normal tissue samples using Wilcoxon's rank-sum tests for all cancer types that had at least ten noncancerous samples from the related tissues. We retrieved the PAM50 intrinsic subtype[72] data of breast cancer samples from Ciriello et al.[43]. We obtained the clinical information of the patients, including pathological stage, vital status, and survival time from TCGA data portal. As different pathological stage terms were provided for different cancer types or even within the same cancer type, we merged them into the same major stage groups: stage I (stage I, stage IA, stage IB), stage II (stage II, stage IIA, stage IIB), stage III (stage III, stage IIIA, stage IIIB, stage IIIC), and stage IV (stage IV, stage IVA, stage IVB, stage IVC). Skin cutaneous carcinoma was excluded from this analysis by stage group since most such samples were not from primary tumors[73].

We assessed the association of ECPA with pathological stage using Spearman's rank correlation. The survival time of patients used in the analysis was the number of days until death or until the last follow-up for patients who were still alive at the time of censoring. We assessed the association of ECPA$_{cell}$ with patient survival times using log-rank tests (patients were split into two groups based on the median ECPA$_{cell}$ value) or the univariate Cox proportional hazards model with the survival package[74]. We performed the analysis in 33 cancer types. Due to the limited sample size and shorter follow-up time, the analysis for some cancer cohorts might have had low statistical power to detect significant correlations. Therefore, we focused on 17 cancer types that had ≥ 75 cases and ≥ 25% events (Fig. 4c, d, Supplementary Fig. 14c and Supplementary Fig. 15c). We used multivariable Cox proportional models (survival ~ stage + ECPA, survival ~ age + stage + ECPA) to assess the additional prognostic power of ECPA$_{cell}$. To evaluate statistical significance, we randomly shuffled the sample labels within each cancer type 1000 times and repeated the analyses to infer the background distribution. The significance of the observed cancer types associated with patient survival ($P < 0.05$ in the log-rank test or univariate Cox model or in both tests with the same direction) was calculated based on the background distribution. We performed a similar analysis by stratifying patients based on the expression level of each gene. Besides using all the expressed genes, we focused on only the therapeutic targets or biomarker genes[46]. All the analyses mentioned above were performed with the Y20, B20, and H11 metrics separately.

**Analysis of single-cell RNA-Seq data**. The processed single-cell RNA-Seq data and the classification of cell types were obtained from Gene Expression Omnibus (GEO) under accession GSE72056 for the melanoma dataset[44] and from figshare (figshare.com/s/711d3fb2bd3288c8483a) for the ovarian cancer ascites dataset[45]. For both datasets, the gene expression levels of each cell were quantified as transcript per million by the original studies and directly used to compute $ECPA_{cell}$ values. The differences in $ECPA_{cell}$ between different cell types in each dataset were compared with Wilcoxon's rank-sum tests.

**Analysis of experimental evolution of xenograft tumor**. The experimental evolution of xenograft tumor was described previously[49]. For MCF10A-HRAS, XT1, XT2, XT3, XT4, XT5, XT6, XT7, and XT8, and the two metastatic tumors, XT8_M1 and XT8_M2, the Poly(A)+mRNA sequences were downloaded from the Sequence Read Archive (accession number PRJNA268433). Based on the gene RPKM values, we calculated $ECPA_{cell}$ values for the nine primary tumor samples and conducted linear regression of $ECPA_{cell}$ against the generation number of the eight derived primary tumor samples (XT1 to XT8).

**Computational simulation of ECPA-based cancer cell evolution**. The evolution of the cancer cell population was simulated with an initial population size $N(0) = 10,000$ cells. The growth of the population follows a Gompertz growth function so that the population size at generation $g$ is $N(g) = N(0) \cdot e^{\frac{\alpha}{\beta}\left(1 - e^{-\frac{22}{24}\beta g}\right)}$, where $\alpha$ is the initial proliferation rate and $\beta$ is the rate of exponential decay of this proliferation rate. The experimentally fitted parameters are $\alpha = 0.56$ and $\beta = 0.0719$ for cancer cell growth per day[75]. The growth time (day) was converted to the number of generations in this study (22 h for a cell cycle duration).

The initial ECPA for each cell was set to 143 (based on the mean $ECPA_{cell}$ of all the TCGA samples), and the optimal ECPA was arbitrarily set at 140 based on the bottom 10% quantile of $ECPA_{cell}$ for all the TCGA samples (we also used other quantile values and observed similar patterns). At each generation, the fitness ($f$) of a cell is $f = e^{-s\left|\frac{ECPA_{cell} - ECPA_{opt}}{ECPA_{opt}}\right|}$, where $s$ (set at 0.01, 0.1, 0.2, 0.5 and 1.0) is the model selection strength on ECPA.

The cell population in generation $g$ was sampled to generation $g+1$ based on cellular fitness given a selective coefficient $s$. In each generation, the ECPA of a cell $k$, $ECPA_{g,k}$ has a probability $v$ ($10^{-6} - 10^{-2}$) of mutating to a value $ECPA'_{g,k}$. $ECPA'_{g,k}$ follows a gamma distribution with mean equal to $ECPA_{g,k}$ and variance equal to 3.12 (calculated based on ECPA of all TCGA samples, except for liver cancer because the ECPA of these samples is much higher than that of the others). Each simulation process was replicated 200 times.

**Analysis of gene categories dysregulated in tumors**. The list of cancer driver genes was taken from Vogelstein et al[4]., and the list of tumor suppressors was from TSGene database (https://bioinfo.uth.edu/TSGene/). Annotation for genes related to AA biosynthesis and transport was downloaded from Molecular Signature Database GO gene sets (http://software.broadinstitute.org/gsea/msigdb/). The list of 530 proliferation-related genes whose expression are significantly positively associated with growth rates was obtained from Waldman et al.[52]. For each cancer type that has at least ten normal samples in TCGA datasets, the normalized counts (or normalized RPKM for LAML and STAD) of genes were averaged for tumor samples and normal tissue samples, respectively. Genes with average RPKM < 1 (for STAD and LAML) or average normalized read count < 20 (for other cancer types) in tumor or normal tissue samples were excluded. Wilcoxon's signed-rank tests were conducted to test whether there is a significant difference in the mean expression levels of genes in each of the four categories (cancer driver genes, tumor suppressors, and genes related to AA biosynthesis or transport) between tumor and normal tissue samples in this cancer type. We also excluded genes in each of the four categories and repeated the pan-cancer analysis of ECPA with the remaining genes.

**Analysis of genes and pathways correlated with ECPA**. To identify pathways enriched in genes with high $ECPA_{gene}$ or low $ECPA_{gene}$, we ranked all the human protein-coding genes based on decreasing $ECPA_{gene}$ and performed gene-set enrichment analyses for the top 6000 genes with highest $ECPA_{gene}$ or the bottom 6000 genes with lowest $ECPA_{gene}$ using DAVID (https://david.ncifcrf.gov/).

To identify genes whose expression levels were associated with $ECPA_{cell}$, we calculated Spearman's rank correlation between $ECPA_{cell}$ and the normalized expression level of each gene in each of the 31 cancer types that have at least 50 samples available. In each cancer type, genes with normalized read count < 20 were excluded from the correlation analysis. Many positively or negatively correlated genes (false discovery rate-adjusted $P$-value < 0.05) are presented in Supplementary Table 8. To identify the gene sets over-represented in positively or negatively correlated genes, we focused on the 20 cancer types that have lower $ECPA_{cell}$ in tumors or have $ECPA_{cell}$ associated with the pathological stage of tumors or patient survival time (Fig. 4), and performed gene-set enrichment analysis with DAVID for genes that had expression levels that correlated with $ECPA_{cell}$ among samples in the same direction in at least 9 of the 20 cancer types. Positively correlated genes and negatively correlated genes were analyzed separately. Wilcoxon's rank-sum tests were conducted to compare the $ECPA_{gene}$ of positively or negatively correlated genes in each over-represented pathway to that of the genomic background.

**Analysis of ECPA with tumor response in anti-PD-1 treatment**. We obtained the patients' treatment response data and the normalized gene expression data from Hugo et al.[56]. We used a one-sided $t$-test to assess whether the $ECPA_{cell}$ values of the responding group were significantly higher than those of the non-responding group. To further assess the statistical significance of the observed $ECPA_{cell}$ difference, we shuffled the biosynthetic costs of 20 AAs 1000 times and repeated the analysis. The empirical $P$-value of the true $ECPA_{cell}$ difference was calculated by the number of permutations with a more significant $P$-value and a larger fold difference in $ECPA_{cell}$ (responding/non-responding) than the true observation. To examine whether $ECPA_{cell}$ can improve the predictive power of clinical variables, we performed model construction using random forests[58] with leave-one-out cross-validation. We considered four groups of candidate features: (i) clinical variables (gender, age, pretreatment, and pathologic stage); (ii) mutation status of the three melanoma driver genes (BRAF, NRAS, and NF1); (iii) mutation load (the number of non-synonymous mutations per patient); and (iv) $ECPA_{cell}$. We first built models using each of the first three feature sets or their combination and then included $ECPA_{cell}$ as an additional feature. We examined the improvement in predictive power between models with and without $ECPA_{cell}$ using a paired $t$-test.

**Processing of ribosome profiling data**. The ribosome profiling data for kidney tumors (six samples of normal and ten samples of tumor kidney tissues) was downloaded from GEO under accession GSE59821[65]. The next-generation sequencing reads were mapped to hg19 using hisat2 (https://ccb.jhu.edu/software/hisat2/index.shtml) based on the genome annotation from ENSEMBL (www.ensembl.org). In each sample, the reads mapped to coding sequence (CDS) region of protein-coding genes were counted using HTSeq-count (https://github.com/simon-anders/htseq) with the parameter "-i gene_id -t CDS", and the RPKM value for each gene was calculated as $n/L/N \times 10^9$, where $n$ is total reads uniquely mapped to CDS region of that gene, $L$ (nt) is the CDS length of longest transcript of that gene, and $N$ is the total number of reads uniquely mapped to protein-coding genes in this library.

**Code availability**. No software was used for data collection. The following software was used to analyze data in this study: R statistical software (v3.3), survival R package (v2.39), bowtie2 (v2.2.1), DAVID (v6.7), hisat2 (v2.0.4), and HTSeq-count (v0.6.1). Custom scripts used in this study are available upon request.

## Data availability

The data that support the findings of this study are available from the corresponding authors upon reasonable request.

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

## Acknowledgements

We thank Drs Chung-I Wu, Wen-Hsiung Li, Hong Wu, Zemin Zhang, Wen Wang, Fuchou Tang, Zhenglong Gu, Gordon Mills, and Yuelin Liu for helpful suggestions. This work was supported by grants from the National Natural Science Foundation of China (Number 91731301) to J.L. and (Number 91731302) to X.L.H.; grants from the U.S. National Institutes of Health (CA175486, CA209851, and CCSG grant CA016672), a

grant from the Cancer Prevention and Research Institute of Texas (RP140462), a University of Texas System STARS award, and the Lorraine Dell Program in Bioinformatics for Personalization of Cancer Medicine to H.L. J.L. is also supported by the grant from the Peking-Tsinghua Center for Life Sciences, and Y.W. is supported by a grant from the Chinese Initiative Postdocs Supporting Program. We thank LeeAnn Chastain for editorial assistance.

## Author contributions

H.L. and J. Lu supervised the whole project and conceived of and designed the research. Hong Zhang, Y.W., J. Li, Huiwen Zhang, H.L., and J. Lu contributed to the data analysis. H.C. and X.H. contributed to the xenograft tumor mRNA-Seq data and conducted the relevant analysis. Hong Zhang, Y.W., J. Li, H.L, and J. Lu wrote the manuscript with input from the other authors.

## Additional information

**Competing interests:** H.L. is a shareholder and on the Scientific Advisory Board for Precision Scientific Ltd. and Eagle Nebula Inc. And all authors declare no other competing interests.

