## [Peer Review File · Nature Communications]

Reviewers' comments:

Reviewer #1 (Remarks to the Author):

The paper by Zhang et al. presents a measure for the energetic cost of synthesizing amino acids (ECPA). Analyzing gene expression data sets from The Cancer Genome Atlas, they show that cancer cells evolve to minimize the total energetic cost associated with amino acid utilization. ECPA is shown to provide a robust prognostic power across cancers. An experimental study of the gene expression of cancer cells throughout different stages of evolution further supports the proposed energy optimization claim.

The novelty of the first section on "The biosynthetic cost underlies the usage of AAs in human proteomes" is moderate, considering that amino acid cost-utilization anti-correlation were demonstrated before for more than a hundred genomics (as noted by the authors). Here, this was extended to more than 10,000 genomes.

The more intriguing part of the paper is with regard to the utilization TCGA data sets, showing significant anti-correlations between the amino-acid biosynthetic cost of genes (ECPA_gene) and their expression level – with stronger anti-correlations found in tumors versus in normal cells.

ECPA_gene is defined as the average cost per amino acid in a gene. It is not clear why normalize by the number of amino acids rather than computing the total energetic cost of amino acids per gene. One could expect that aiming to minimize the total energetic cost, cancer cells would prefer expressing genes with a lower total cost (rather than genes whose average cost per AA is low). This needs to be further tested.

Overall, the paper is nicely written and could be of interest within the cancer metabolic research community. Statistical analysis is valid.

Reviewer #2 (Remarks to the Author):

The authors investigate the association between the energetic cost of amino acid synthesis and cancer evolution. While I find this work quite interesting and I enjoyed reading it, there are some issues that need to be evaluated regarding the interplay between association and causality at the point of drawing conclusions for the reported data. Specifically, the following points need further consideration:

1- The authors report that genes that are overexpressed in cancer are associated with a lower energetic cost of synthesis, lower ECPA_{gene}. However, the authors do not investigate whether such association can be explained by a lower ECPA for genes involved in proliferation together with the fact that proliferation is increased in cancer. In fact, the impact of growth on the evolution of amino acid usage may be more general. We would expect that proteins required during proliferation should have lower ECPA, because they need to be synthesized continuously as cells grow. In contrast, proteins that are required during quiescent states are synthesized at much lower rate to balance the small basal rate of protein turnover. Therefore, the authors need to test for the association between gene correlation with proliferation and ECPA_{gene}. The association between the expression of a gene with proliferation could be quantified, for example, by using gene expression signatures of proliferation and quantifying the association between the signature intensity and the gene expression.

2- If the point raised above turns out to be true, that genes expressed during proliferation have significantly higher ECPA_{gene} than those expressed during quiescent or non-proliferative states, then there are many associations that need to be revised in the light of this evidence. In other words, many associations reported by the authors would be explained through the association of ECPA_{gene} with proliferation.

2A Is the tendency to have low ECPA in genes up-regulated in cancer significantly above of what expected from the up-regulation of genes associated with proliferation and the negative association between genes involved in proliferation and ECPA?

2B If the samples in Fig. 4D are stratified by a proliferation signature and ECPAcell, is ECPAcell still predictive of survival? In other words, is ECPAcell and prognostic factor independent from proliferation?

2C Is the tendency of decreased ECPAcell with serial tumor transplantations (Fig. 5A) independent of an increase in the degree of proliferation between serial transplantations and a negative correlation between ECPAcell and proliferation?

3- One should make a distinction between the evolutionary pressures experienced by organisms and the evolutionary pressures experienced by a tumor within an organism that is trying to maintain homeostasis. In that sense, it is not obvious that there should be a selection pressure to use non-essential amino acids with low ECPA if the host is taking care of the cost of making them available. As noted above, the observation of low ECPA in cancer may just be a consequence of increased proliferation. Yet, it is possible that the concentration of non-essential amino acids in plasma is dictated by their ECPA and therefore there is indeed a selective pressure to use them in such proportions. Can the authors report what is the relationship between amino acids ECPA and their concentration in human plasma?

Minor comments

4- The authors should provide better of ECPAcell. I'm not sure I truly understand what is the meaning of ECPAcell, which is a central concept of their work.

5- The subsection title "ECPAcell is an essential cellular property..." should be changed. There no probe of essentially in that subsection. At most an association.

Reviewer #3 (Remarks to the Author):

In their manuscript Zhang et al first described a universal trend that amino acid costs anti-correlate with their proteomic occurrences in over 10,000 species. Then they showed that in cancer cells / samples that grow much faster, the anti-correlations are often stronger than in normal tissues, suggesting selection for energy efficient usage is a key feature of rapidly-proliferated cells such as cancers. They introduced ECPA indexes to quantify the use of the 20AA at gene and cell/sample levels and found that they are excellent indicators for expression abundances, cell proliferation rate and even patient prognosis outcomes.

Overall the manuscript is very well written. The language was surprisingly good for non-native speakers, without any obvious flaws. The study and results were described in necessary details with concise and professional language. The logic was clear, and the conclusions were well supported by their data.

The results are quite interesting and useful, although in my personal opinion the potential clinical utility of this study can be limited: it is a bit impractical to use the expression abundance of all genes of a patient to predict his/her clinical outcome (in contrast to just a few marker genes). But in any case, it is good to have a novel and independent marker for prognosis. Therefore, I would like to recommend it to be published, should the authors could address the following issues.

Major issues:

1. If I understand correctly the AA costs used in this study were calculated under aerobic

conditions. However, it has been shown that many cancers have low levels of oxygen and acquire needed energy in great part by fermentation. I thus wonder if the conclusions hold true if the authors use AA costs calculated under anaerobic conditions.

Minor issues:

1. If the selection for energy efficiency at protein level drives the preferable use of cheaper amino acids, we would expect that the anti-correlation between the ECPA-gene and the *protein* expression abundances to be stronger, or at least as strong as the anti-correlation between the ECPA-gene and *mRNA* expression abundances. However, the opposite was found as indicated in Figure 3C. Can the authors provide some explanation for this?

Reviewer #4 (Remarks to the Author):

What are the major claims of the paper?

1. The authors have developed a novel metric of amino acid biosynthetic cost (energy cost per amino acid - ECPA) to characterize the use of amino acids in proteins synthesis. This metric was then compared with with gene expression and proteomic data to investigate changes in amino acid utilization across a range of cancer cell models.

2. On the basis of relatively small changes in ECPA between normal and cancer cells, the authors claim that cancer cells evolve optimized gene expression profiles to utilize amino acids more economically. The authors claim that this effect may be a common principle in cancer evolution.

3. With respect to potential clinical utility of the findings, the authors claim that ECPA provides generalizable prognostic power across many cancer types, and utility in predicting response to immunotherapy.

Are they novel and will they be of interest to others in the community and the wider field?

4. There is currently intense interest in dysregulated amino acid metabolism in cancer biology, and it's potential exploitation as a therapeutic target. The ECPA metric is a novel attempt to quantify one aspect of amino acid metabolism, namely biosynthetic cost.

5. However, this study relies on number of incorrect assumptions about amino acid metabolism in cancer cells, significantly limiting its likely impact on the field. Further, while changes in ECPA in tumours reach statistical significance, the magnitude of these effects are relatively small, bringing their biological or functional relevance under question.

6. Significant limitations in interpretation of data in the context of contemporary thinking in cancer biology and clinical management of disease lessen the potential interest and utility of this manuscript to the cancer metabolism field, and cancer biology more broadly.

Is the work convincing, and if not, what further evidence would be required to strengthen the conclusions?

7. The manuscript is based on an overly simplistic assumption that de novo amino acid synthesis is the predominant source of amino acids for protein synthesis in cancer cells. This ignores a huge and rapidly evolving body of evidence about both the source and use of amino acid in cancer cells. Significant sources of amino acids in cancer cell include amino acid import, recycling via autophagy, and protein degradation by the proteasome. Similarly, the metabolic fate of amino acids extends beyond protein synthesis to include biosynthesis of nucleotides, fatty acids, reductive intermediates, etc. In other words, what is the relative importance/contribution of amino acid biosynthesis to cancer cell proliferation?

8. Overall, the biological interpretations and conclusions are not supported by the experimental data and are highly speculative. The concept that amino acid biosynthetic cost plays a major role in tumour biology and evolution needs to be tested experimentally (e.g. measure changes in the relative contribution of individual AA in various metabolic pathways like protein, lipid, nucleotide synthesis, or demonstrate functional changes in tumour or cell behavior following inhibition or manipulation of de novo amino acid biosynthesis).

9. Have the authors excluded the possibility that the observed small changes in ECPA are a proxy

for some other effect such as proliferation?

10. Further evidence supporting the conclusions would also need to include consideration of tumour subtypes and comparison of the putative prognostic value of ECPA against multiple gene expression panels.

11. Altered gene expression is only one aspect reflecting underlying functional and genetic changes in cancer cells. These cells also carry a significant mutational load. What is the potential contribution of mutations (particularly those that affect protein stability and turnover through altered post-translational modifications) on energy cost of biosynthesis?

12. P9: Regarding the statement that "ECPAcell is an essential cellular property in cancer evolution" - Gene expression data used in these calculations is largely collected at the tissue/tumour level and does not provide single cell resolution. In this context, ECPAcell is a somewhat misleading term, as energy cost has not been calculated at the single cell level. This also ignores the contribution of stromal and immune cell populations to gene expression signatures. These can be a significant component of tumours (e.g. in pancreatic cancer). Single cell transcriptomes are now available for a number of cancer types and analysis of these data in the context of ECPAcell would be much more informative, and relevant to tumour biology and evolution.

13. P8: Please provide evidence for limited availability of AAs in the tumour microenvironment?

14. P10: "strongly suggest that ECPAcell is selected for reduction during tumour evolution... shaped the evolutionary roadmap". This claim is somewhat hyperbolic given the very small effect measured.

On a more subjective note, do you feel that the paper will influence thinking in the field?

15. As the paper doesn't reflect or incorporate several aspects of contemporary thinking in cancer cell metabolism, cellular heterogeneity, tumour subtype, etc., its potential to influence thinking in the field is limited.

Further questions and concerns about the paper.

16. Throughout the paper, gene expression measurements used were collected at the tissue (i.e. tumour) level and represent the "average" expression of a gene across all cells present in the sample (including stroma), but interpretation and discussion often refers to single cell level effects. For example, on page 4 "Based on ECPAcell and the overall gene expression profile of a cell...". This is a significant error that must be addressed.

17. P8: It is important to consider the critical role of heterogeneity in gene expression profiles between individual patients. Equally, and using the same logic, heterogeneity of gene expression between individual cells within a tumour is a very important consideration that has not been measured in this analysis (see comments above)

18. P9, re: "superiority of ECPAcell over individual genes as a potential prognostic marker". The data presented do not support this claim as the analysis ignores the important distinction of cancer subtypes based on cell and genetic markers, which are an important part of clinical decision making on treatment and prognosis). Further, as single gene markers are rarely used, the clinical utility of this claim is questionable in contemporary management of cancer.

19. P11: re low ECPAcell as a marker of resistance to immunotherapies. How does this, measure compare with PD1/PDL1 expression?

20. P12: The claim about the predictive power of ECPAcell was not actually tested in a predictive context, it is based on a single cohort and should be tested against other models and cohorts.

21. P12: The claim that "the use of all 20 AAs in human proteomes is constrained by their synthetic costs" is rather broad and must be supported by cited references.

22. P12: re claim that cancer cells evolve to use AAs more efficiently was not actually tested as measurements were made at the cellular level.

23. P13: Individual genes are increasingly rarely used as prognostic markers, so comparing the power of ECPAcell against these is somewhat outdated. Further, many tumour types are increasingly being parsed into subtypes based on gene expression or mutational signatures for diagnostic, prognostic and therapeutic use. The performance of ECPAcell as a prognostic marker needs to be evaluated in this context.

24. P13/14: How does ECPAcell factor in the cost of AA acquisition from the environment, which represents a significant source of AAs in tumour cells? In this context, wouldn't there be a bias in tumour cell requirements for biosynthesis of NEAA?
25. P13: How do ECPAcell values compare with protein synthesis? This can be measured experimentally in the context of inhibition of protein synthesis.
26. Fig1F regression does not look correct. There are 2 clear clusters in the data?
27. Fig3C: need to see individual correlations, not just bar plots.
28. Fig 3D: Scales of graphs are misleading. Magnitude of effect is very small here.
29. Fig4A graph scale condensed, magnitude of effect is very small. Need to split cancers by subtype.
30. Fig5a: regression fits are not convincing. Appears to be an initial increase in ECPAcell, followed by a subsequent decrease. Again, magnitude of effect is very small.
31. Fig 7a: again, magnitude of effect is very small.

Point-to-point response

Reviewer #1

The paper by Zhang et al. presents a measure for the energetic cost of synthesizing amino acids (ECPA). Analyzing gene expression data sets from The Cancer Genome Atlas, they show that cancer cells evolve to minimize the total energetic cost associated with amino acid utilization. ECPA is shown to provide a robust prognostic power across cancers. An experimental study of the gene expression of cancer cells throughout different stages of evolution further supports the proposed energy optimization claim.

The novelty of the first section on “The biosynthetic cost underlies the usage of AAs in human proteomes” is moderate, considering that amino acid cost-utilization anti-correlation were demonstrated before for more than a hundred genomics (as noted by the authors). Here, this was extended to more than 10,000 genomes.

The more intriguing part of the paper is with regard to the utilization TCGA data sets, showing significant anti-correlations between the amino-acid biosynthetic cost of genes (ECPA_{gene}) and their expression level – with stronger anti-correlations found in tumors versus in normal cells.

ECPA_{gene} is defined as the average cost per amino acid in a gene. It is not clear why normalize by the number of amino acids rather than computing the total energetic cost of amino acids per gene. One could expect that aiming to minimize the total energetic cost, cancer cells would prefer expressing genes with a lower total cost (rather than genes whose average cost per AA is low). This needs to be further tested.

Overall, the paper is nicely written and could be of interest within the cancer metabolic research community. Statistical analysis is valid.

Response: We greatly appreciate the favorable review from this reviewer. This reviewer well summarized the findings of our study.

In our previous submission, we calculated the average energetic cost per amino acid for a protein (ECPA_{gene}) or for the whole transcriptome (ECPA_{cell}). The reviewer suggests us to examine whether cancer cells prefer expressing genes with a lower total cost rather than genes whose average cost per AA is low. In this revision, we followed the reviewer’s suggestion and performed new analyses to address this issue. We first calculated the total energetic cost of each gene (EC_{gene}) as the sum of the biosynthetic costs of amino acids (AAs) in the peptide sequence. We found that in both normal and tumor tissues, the EC_{gene} showed significant negative correlations with gene expression levels (Supplementary Fig. 32a), indicating that cells prefer expressing genes with lower total energetic costs. Compared to genes that are significantly down-regulated, genes that are significantly up-regulated in cancer cells tend to have significantly lower EC_{gene} (Supplementary Fig. 32b). We also calculated the EC_{cell}, which is the average EC_{gene} of genes weighted by their expression levels in the transcriptomes. The EC_{cell} values are significantly lower in tumors than in normal tissues (Supplementary Fig. 33). These results collectively suggest that cancer cells prefer expressing genes with lower total costs. However, when we performed the pan-cancer analyses with the EC_{cell} values, we did not find the EC_{cell} show a consistent association with the pathological stage of tumors or survival time of patients (Supplementary Fig. 33). Therefore, we propose that ECPA_{cell} is more powerful than EC_{cell} as a prognostic and predictive marker for cancer progression.

We present the new results in Line 316–327 of Page 12 as follows:

“Cancer cells might prefer expressing proteins with lower total biosynthetic cost (rather than protein whose average cost per AA is low). To evaluate this possibility, we calculated the total energetic cost of each protein (EC_{gene}) as the sum of the biosynthetic costs of AAs in the protein sequence. As expected, genes with higher EC_{gene} have lower abundance in both normal tissues and tumors (Supplementary Fig. 32a). Compared to genes that are significantly down-regulated, genes that are significantly up-regulated in cancer cells tend to have significantly lower EC_{gene} (Supplementary Fig. 32b). Moreover, the EC_{cell} values, which are calculated

as average EC_{gene} of genes weighted by their expression levels (Methods), are significantly lower in tumors than in normal tissues (Supplementary Fig. 33). These results suggest that cancer cells prefer expressing genes with lower total costs. However, in the pan-cancer analysis, the pathological stage of tumors or survival time of patients is generally not associated with the EC_{cell} parameters (Supplementary Fig. 33), suggesting that $ECPA_{cell}$ is more powerful than $ECPA_{gene}$ as a prognostic and predictive marker for cancer progression.”

Reviewer #2

The authors investigate the association between the energetic cost of amino acid synthesis and cancer evolution. While I find this work quite interesting and I enjoyed reading it, there are some issues that need to be evaluated regarding the interplay between association and causality at the point of drawing conclusions for the reported data.

Response: Thank you very much for your enthusiasm for our work and your favorite reviews. We greatly appreciate your insightful comments and suggestions. The major concern of this reviewer is whether the reduced $ECPA_{cell}$ values in the cancer cells are mainly caused by changes in expression level of proliferation-related genes. In this revision, we performed new analyses to address your concern. We found that the association between changes in $ECPA_{cell}$ and cancer progression in the pan-cancer analyses we observed is not likely caused by the changes in proliferation-related genes alone. Please refer to the follow point-to-point response for details.

Specifically, the following points need further consideration:

1- The authors report that genes that are overexpressed in cancer are associated with a lower energetic cost of synthesis, lower $ECPA_{gene}$. However, the authors do not investigate whether such association can be explained by a lower $ECPA$ for genes involved in proliferation together with the fact that proliferation is increased in cancer. In fact, the impact of growth on the evolution of amino acid usage may be more general. We would expect that proteins required during proliferation should have lower $ECPA$, because they need to be synthesized continuously as cells grow. In contrast, proteins that are required during quiescent states are synthesized at much lower rate to balance the small basal rate of protein turnover. Therefore, the authors need to test for the association between gene correlation with proliferation and $ECPA_{gene}$. The association between the expression of a gene with proliferation could be quantified, for example, by using gene expression signatures of proliferation and quantifying the association between the signature intensity and the gene expression.

Response: Thank you for raising this concern. In this revision, we compiled a list of 530 proliferation-related genes from a recent study (Waldman et al., 2013). These proliferation-related genes were identified because their expression levels showed significant positive association with the growth rate of the cancer cells among the NCI-60 cancer cell lines. We found these proliferation-related genes tend to be up-regulated in cancer cells compared to normal tissue (Supplementary Fig. 28a). Consistent with this reviewer’s expectation, the proliferation-related genes do have lower $ECPA_{gene}$ than the remaining genes (Supplementary Fig. 28b). However, the association between $ECPA_{cell}$ and the cancer progression in the pan-cancer analysis remains intact when we excluded the proliferation-related genes from the analysis (Supplementary Fig. 29). The reduction of $ECPA_{cell}$ in experimental evolution of xenograft tumors is not affected either after excluding proliferation-related genes (Supplementary Fig. 31). Furthermore, the $ECPA_{cell}$ values calculated with only the proliferation-related genes are not significantly associated with the patient survival time in most cancer types (Supplementary Fig. 30). These results collectively suggest that the whole patterns we observed are unlikely caused by the changes in proliferation-related genes alone.

We report the updated results in Line 308–316 of Page 12 as follows:

“The expression levels of proliferation-related genes⁷⁵ are increased in tumors compared to the matched normal samples (Supplementary Fig. 28a). Although the proliferation-related genes have lower $ECPA_{gene}$ than the remaining genes (Supplementary Fig. 28b), the results of pan-cancer analyses are only slightly

affected after excluding these genes (Supplementary Fig. 29). Also, the $ECPA_{cell}$ values calculated with only the proliferation-related genes do not consistently show association with the patient survival time (Supplementary Fig. 30). Furthermore, the reduction of $ECPA_{cell}$ during experimental evolution of xenograft tumors still holds when the proliferation-related genes were excluded (Supplementary Fig. 31). These results suggest the association between $ECPA_{cell}$ and the cancer progression is unlikely caused by the changes in proliferation-related genes alone.”

2- If the point raised above turns out to be true, that genes expressed during proliferation have significantly higher $ECPA_{gene}$ than those expressed during quiescent or non-proliferative states, then there are many associations that need to be revised in the light of this evidence. In other words, many associations reported by the authors would be explained through the association of $ECPA_{gene}$ with proliferation.

2A Is the tendency to have low $ECPA$ in genes up-regulated in cancer significantly above of what expected from the up-regulation of genes associated with proliferation and the negative association between genes involved in proliferation and $ECPA$?

2B If the samples in Fig. 4D are stratified by a proliferation signature and $ECPA_{cell}$, is $ECPA_{cell}$ still predictive of survival? In other words, is $ECPA_{cell}$ and prognostic factor independent from proliferation?

2C Is the tendency of decreased $ECPA_{cell}$ with serial tumor transplantations (Fig. 5A) independent of an increase in the degree of proliferation between serial transplantations and a negative correlation between $ECPA_{cell}$ and proliferation?

Response: As we described in the point-to-point response to your previous comments, the proliferation-related genes contribute to the reduced $ECPA_{cell}$ values in the cancer cells. However, the association between $ECPA_{cell}$ and cancer pathological stages or patient survival status are still evident after excluding proliferated-related genes (Supplementary Fig. 29). Also, the $ECPA_{cell}$ value calculated with only the proliferation-related genes is not significantly associated with the patient survival time (Supplementary Fig. 30). The tendency of decreased $ECPA_{cell}$ during serial tumor transplantations is unaffected after excluding proliferation-related genes (Supplementary Fig. 31). Altogether, these results suggest the up-regulation of proliferation-related genes during tumor progression cannot fully explain the association of $ECPA_{cell}$ with tumor progression or patient’s survival in the pan-cancer analysis.

3- One should make a distinction between the evolutionary pressures experiences by organisms and the evolutionary pressures experienced by a tumor within an organism that is trying to maintain homeostasis. In that sense, it is not obvious that there should be a selection pressure to use non-essential amino acids with low $ECPA$ if the host is taking care of the cost of making them available. As noted above, the observation of low $ECPA$ in cancer may just be a consequence of increased proliferation. Yet, it is possible that the concentration of non-essential amino acids in plasma is dictated by their $ECPA$ and therefore there is indeed a selective pressure to use them in such proportions. Can the authors report what is the relationship between amino acids $ECPA$ and their concentration in human plasma?

Response: We apologize that we did not put enough emphasis on this point in our previous submission. Tumorigenesis is an evolutionary process and the populations of cancerous cells evolve like populations of organisms (Cairns, 1975; Korolev et al., 2014; McGranahan and Swanton, 2017; Merlo et al., 2006; Nowell, 1976; Wu et al., 2016). Although the host is trying to maintain the homeostasis under normal conditions, supplying enough AAs for tumor growth does no good to itself and will only accelerate its elimination. Therefore, it is reasonable that tumors will utilize AAs more economically to support elevated proliferation rate. Cancer cells with lower $ECPA_{cell}$ would gain a slight selective advantage compared to the other cells in tumor microenvironment as free AAs of lower biosynthetic costs are more abundant in that environment. In support of this view, the concentration of AAs in human plasma showed strong anticorrelations with their cost. The relationship between energy cost of AAs and their concentration in the plasma has been reported in Supplementary Fig. 8 of our original submission. This result was reported in Line 157–160 of Page 7 in our initial submission as follows:

“Previous results suggest cancer patients usually have dysregulated AA levels in blood⁵⁴⁻⁶⁴ or tumor tissues^{57,58}. However, we observed similar negative correlations between the cost and abundances of the free AAs in tumor and matched normal tissues for a variety of cancer types (Supplemental Fig.S8 and Supplemental Table S7).”

In this revision, we also discussed the difference in the evolutionary pressures acting on the organism level and the pressure on tumor cells (Line 398–403 of Page 15) with the following sentences:

“Tumorigenesis is an evolutionary process and the populations of cancerous cells evolve like populations of organisms¹⁻⁶. Noteworthy, the selective pressure acting at the organism level is different from that experienced by the cancer cells. While human normal cells are required to maintain AA homeostasis for normal biological function, cancer cells with lower $ECPA_{cell}$ would gain the selective advantage over the other cells in tumor microenvironment as free AAs of lower biosynthetic costs are more abundant in that environment.”

Minor comments

4- The authors should provide better of $ECPA_{cell}$. I’m not sure I truly understand what is the meaning of $ECPA_{cell}$, which is a central concept of their work.

Response: Thanks for this suggestion. Besides Fig. 3b, in this revision, we also briefly illustrate the calculation of $ECPA_{cell}$ in Line 189–191 of Page 8:

“Of note, $ECPA_{cell}$, which measures the virtual average cost of AAs in the proteomes, not only considers the compositions of AAs in the protein sequences, but also incorporates the relative abundance of the proteins in the cells.”

5- The subsection title “ $ECPA_{cell}$ is an essential cellular property...” should be changed. There no probe of essentially in that subsection. At most an association.

Response: Thanks for pointing this out. We rephrased this subtitle with the sentence “ **$ECPA_{cell}$ is reduced in the experimental evolution of xenograft tumors**” in this revision. (Line 258 of Page 10).

Reviewer #3

In their manuscript Zhang et al first described a universal trend that amino acid costs anti-correlate with their proteomic occurrences in over 10,000 species. Then they showed that in cancer cells / samples that grow much faster, the anti-correlations are often stronger than in normal tissues, suggesting selection for energy efficient usage is a key feature of rapidly-proliferated cells such as cancers. They introduced $ECPA$ indexes to quantify the use of the 20AA at gene and cell/sample levels and found that they are excellent indicators for expression abundances, cell proliferation rate and even patient prognosis outcomes.

Overall the manuscript is very well written. The language was surprisingly good for non-native speakers, without any obvious flaws. The study and results were described in necessary details with concise and professional language. The logic was clear, and the conclusions were well supported by their data.

The results are quite interesting and useful, although in my personal opinion the potential clinical utility of this study can be limited: it is a bit impractical to use the expression abundance of all genes of a patient to predict his/her clinical outcome (in contrast to just a few maker genes). But in any case, it is good to have a novel and independent maker for prognosis. Therefore, I would like to recommend it to be published, should the authors could address the following issues.

Response: We greatly appreciate the enthusiasm and the positive feedback from this reviewer. The comments and suggestions are precious and very helpful for us to make this revision. In this revision,

we have fully considered your comments and made the revisions accordingly. Please refer to the point-to-point response for details.

Major issues:

1. If I understand correctly the AA costs used in this study were calculated under aerobic conditions. However, it has been shown that many cancers have low levels of oxygen and acquire needed energy in great part by fermentation. I thus wonder if the conclusions hold true if the authors use AA costs calculated under anaerobic conditions.

Response: This question is pertinent. This reviewer raised the concern that tumors often experience hypoxia and obtain part of cellular energy via fermentation and the costs of AAs might be different under anaerobic condition. Here, we repeated the pan-cancer association analysis with the anaerobic costs of AAs and found our conclusions still hold if the anaerobic costs of AAs are used in the analysis (Supplementary Figs. 21-23). It should be noted that although we reproduced the similar results with anaerobic costs of AAs, cancer cells obtained most of AAs from exogenous supply rather than de novo synthesis. The results might be caused by the strong correlation between the anaerobic and aerobic costs for each AA. We reported the updated analysis in Line 244–253 of Page 14 with the following sentences:

“Noteworthy, in the above analyses the biosynthetic cost of AAs was calculated under aerobic conditions. Tumors often experience hypoxia⁷⁰ and obtain part of cellular energy via fermentation⁷¹. Hence, we repeated the pan-cancer association analysis with anaerobic costs of AAs and found our conclusions still hold (Supplementary Figs. 21-23). We should be cautious that these patterns do not necessarily suggest that cancer cells synthesize all the AAs under anaerobic conditions because cancer cells obtained most AAs from exogenous supplies. Moreover, these patterns might also be caused by the strong correlation between the cost of an AA under the anaerobic and aerobic conditions.”

Minor issues:

1. If the selection for energy efficiency at protein level drives the preferable use of cheaper amino acids, we would expect that the anti-correlation between the ECPA-gene and the *protein* expression abundances to be stronger, or at least as strong as the anti-correlation between the ECPA-gene and *mRNA* expression abundances. However, the opposite was found as indicated in Figure 3C. Can the authors provide some explanation for this?

Response: Thank you for raising this concern. We fully agree with this reviewer on this point. We think the observed weaker correlation between ECPA_{gene} and protein abundance than that between ECPA_{gene} and mRNA abundance might be caused by the technical limitation in quantifying protein abundance. Only a limited number of genes are sampled in a typical mass spectrometry measurement and the reproducibility of proteomic quantification is much lower than mRNA-Seq. In this revision, we explained this discrepancy in Line 157–161 of Page 7 in the revised manuscript as follows:

“Of note, in the above analysis, the correlation between ECPA_{gene} and protein abundance is in general weaker than that between ECPA_{gene} and mRNA abundance (Fig. 3c and Supplementary Fig. 8), presumably because gene expression measured by mRNA-Seq is more comprehensive and accurate than the proteome quantified by mass spectrometry⁵⁹.”

Reviewer #4

General response: We thank this reviewer for the time and efforts in the thorough review of our manuscript. This reviewer raised a series of concerns and critiques of our previous submission. While we found many comments from this reviewer are pertinent and enlightening, we think some of the critiques might stem from the misunderstanding of the ECPA concept we proposed. Briefly, the major concerns of this reviewer can be summarized as follows:

1. This reviewer assumed that our ECPA metrics quantify the actual biosynthetic energy cost of AAs in cancer cells and questioned the validity of ECPA. Based on this assumption, this reviewer had

- the impression that: I) we assumed the *de novo* biosynthesis is the primary sources of AAs in human cancer cells; II) we did not realize that cancer cells can acquire AAs with diverse mechanisms such as importing AAs, recycling via autophagy, or protein turnover; and III) we ignored the fact that AAs also participate in other cellular processes except for protein biosynthesis.
2. We did not consider the heterogeneity of tumors among patients of the same cancer type and the heterogeneity of cells from the same cancer type.
 3. We did exclude the possibility that the changes in ECPA during cancer progression is caused by other factors such as proliferation.
 4. We did not exclude the possibility that DNA mutations in tumor samples would cause changes in protein sequences, which further caused the observed differences in ECPA_{cell} between tumor and normal samples.
 5. This reviewer also raised the concern that the magnitude of the ECPA_{cell} differences between normal and tumor cells is small, and therefore we might have over-stated the prognostic power and potential clinical utility of ECPA.

In this revision, we made the following changes to address the reviewer's concerns:

- 1) We made it clearer that the ECPA_{gene} or ECPA_{cell} metric aims to describe how efficiently human cells manage the given AA pool for protein synthesis. Neither ECPA_{gene} nor ECPA_{cell} measures the actual energy human cells invest to synthesize the AAs endogenously, since the EAAs and most NEAAs in human cells are ultimately taken from the autotrophs. We also emphasized that ECPA_{gene} or ECPA_{cell} can be treated as the average "price-tag" for the AAs in a protein or in the proteomes, respectively.
- 2) We followed the reviewer's suggestions and performed new analyses of tumor subtypes and single-cell RNA-Seq data. Using breast invasive carcinoma (BRCA) as an example, we show that the reduced ECPA_{cell} is common to all tumor subtypes. By analyzing the single-cell RNA-Seq data of melanoma and ovarian carcinoma, we found the ECPA_{cell} of malignant cells is significantly lower than the stromal and immune cells in the same tumor microenvironment.
- 3) We performed new analyses to show that the association between changes in ECPA_{cell} and cancer progression in the pan-cancer analyses we observed is not likely caused by the changes in proliferation-related genes alone.
- 4) We considered the protein sequence changes caused by DNA mutations in the cancer cells and calculated the ECPA_{cell} values. Our new results indicate that AA changes caused by DNA mutations in the cancer cells cannot explain the observed differences in ECPA_{cell} between tumor and normal samples.
- 5) We removed some offending sentences and rephrased our sentences regarding some results and interpretations.

Despite these extensive changes, our previous conclusion remains intact. All the revisions were marked in BLUE in the revised manuscript. We hope the revisions are adequate and satisfactory to this reviewer. Please refer to the following point-to-point response for details.

What are the major claims of the paper?

1. The authors have developed a novel metric of amino acid biosynthetic cost (energy cost per amino acid - ECPA) to characterize the use of amino acids in proteins synthesis. This metric was then compared with with gene expression and proteomic data to investigate changes in amino acid utilization across a range of cancer cell models.
2. On the basis of relatively small changes in ECPA between normal and cancer cells, the authors claim that cancer cells evolve optimized gene expression profiles to utilize amino acids more economically. The authors claim that this effect may be a common principle in cancer evolution.

3. With respect to potential clinical utility of the findings, the authors claim that ECPA provides generalizable prognostic power across many cancer types, and utility in predicting response to immunotherapy.

Response: Thank you for the summary.

Are they novel and will they be of interest to others in the community and the wider field?

4. There is currently intense interest in dysregulated amino acid metabolism in cancer biology, and it's potential exploitation as a therapeutic target. The ECPA metric is a novel attempt to quantify one aspect of amino acid metabolism, namely biosynthetic cost.

Response: We agree with the reviewer that amino acid metabolism is intensely studied in cancer biology (Pavlova and Thompson, 2016; Simińska and Koba, 2016; Tsun and Possemato, 2015; Vander Heiden and DeBerardinis, 2017). As we summarized in Discussion of our previous submission:

“Besides protein synthesis, AAs participate in cellular processes such as nucleotide synthesis, cellular signaling, and regulation of gene expression^{86,87}. It is well recognized that certain AAs, such as asparagine^{37,41,82}, glycine⁴⁰, glutamine^{18,35,36,88,89}, leucine⁹⁰, proline^{91,92}, and serine⁹³, is crucial for the survival and proliferation of cancer cells.”

However, our focus in this study is how the human normal and cancer cells manage the expenditure of AAs in protein synthesis rather than the dysregulation of amino acid metabolism in cancer cells. We wish this reviewer would agree with us that this point is important because recent studies have demonstrated that AAs contribute most to cellular mass and protein synthesis is the most AA-consuming cellular process (Hosios et al., 2016).

As we summarized in the general response to you, the ECPA_{gene} or ECPA_{cell} metrics we developed can be treated as the average “price-tag” for the AAs in a protein or in the proteomes, respectively. Neither parameter measures the actual energy human cells invest to synthesize all the AAs as the EAAs and most NEAAs in human cells are ultimately taken from the autotrophs. Therefore, the ECPA_{gene} or ECPA_{cell} metric aims to describe how human cells manage the utilization of the 20 AAs in the proteome since the relative abundance of AAs is anti-correlated with the biosynthetic cost in autotrophs and heterotrophs. We apologize for not making this message clear enough in the original manuscript. In the revised manuscript, we clarified this point in Introduction (Line 72–77 of Page 4) as follows:

“Since the EAAs and most NEAAs in human cells are ultimately taken from the autotrophs, neither ECPA_{gene} nor ECPA_{cell} measures the actual energy human cells invest to synthesize the AAs endogenously. Rather, these parameters can be treated as the average “price-tag” for the AAs in a protein or the proteomes, respectively. As the relative abundance of AAs is anti-correlated with the biosynthetic cost in the living organisms, the ECPA metrics inform how efficiently human cells manage the given AA pool for protein synthesis.”

5. However, this study relies on number of incorrect assumptions about amino acid metabolism in cancer cells, significantly limiting its likely impact on the field. Further, while changes in ECPA in tumours reach statistical significance, the magnitude of these effects are relatively small, bringing their biological or functional relevance under question.

Response: We are sorry that we have given this reviewer the impression that our analyses relied on incorrect assumptions about AA metabolism in human cells.

In the first part of the manuscript, we analyzed more than 10,000 genomes and found the anti-correlation between the biosynthetic energy cost and utilization of amino acids in all the domains of life (Line 82-107). Our results support the previous hypothesis that in the autotrophs (bacteria, yeast, and plants which can synthesize all 20 proteinogenic AAs), the anticorrelation between the biosynthetic cost and usage of AAs is driven by natural selection for bioenergetic efficiency (Akashi and Gojobori, 2002). Interestingly, although animals can synthesize only 11 NEAAs (Payne and Loomis, 2006), we also observed significant C-U anticorrelations for all the 20 AAs in humans and other animals when the cost

of AA biosynthesis in bacteria (B20) or yeast(Y20) is employed. Since the biosynthetic pathway of NEAAs might be different in humans and other animals compared to yeast or bacteria (Lehninger et al., 2008), we calculated the biosynthetic cost for each NEAA in humans (H11) following previous studies in bacteria (Akashi and Gojobori, 2002) or yeast (Raiford et al., 2008; Wagner, 2005), while taking into account the differences. With the H11 cost metric, we also detected significant C-U anti-correlation of AAs in humans and other animals. Our results are in line with the notion that EAAs and most NEAAs in animal cells are ultimately taken from the autotrophs in which the bioavailability of an AA is constrained by its biosynthetic cost (Heizer et al., 2011; Krick et al., 2014; Swire, 2007).

In this revision, we highlight the difference between autotrophs and heterotrophs with the following sentences (Line 118–124): “Our results support the notion that in autotrophs (bacteria, yeast, and plants), which can synthesize all 20 proteinogenic AAs, the C-U anticorrelation of AAs is driven by natural selection for bioenergetic efficiency⁴². In line with previous discoveries^{45,46,52}, we also observed significant C-U anticorrelations for all the 20 AAs in humans and other animals even though animals can synthesize only 11 NEAAs²⁶. Since EAAs and most NEAAs in animal cells are ultimately taken from the autotrophs in which the bioavailability of an AA is constrained by its biosynthetic cost^{45,46,52}, in Fig. 2a, we illustrate how the biosynthetic energy cost underlies the usage of AAs in human proteomes.”

We also conducted simulations to show that the mixtures of AAs from the two sources: (NEAAs endogenously synthesized in animal cells and all the 20 AAs ultimately taken from autotrophs) yield significantly negative correlations between the overall abundance and cost of all 20 AAs in autotrophs (Supplementary Fig. 6). Moreover, we also compiled previous the experimental data show that the biosynthetic cost is significantly anticorrelated with the abundance of free AAs in human and animal tissues (Fig. 2b).

Altogether, we conclude that the biosynthetic costs of all 20 AAs constrain the relative abundances of free AAs in human cells, which further shapes AA usage in the proteomes (Fig. 2a).

We should point out that in this study, we did not assume that human cancer cells rely on the AAs endogenously synthesized in cancer cells for protein synthesis. Our efforts are summarized as follows:

1) In Introduction (Line 48–55), we clearly indicated that: “Mammalian cells can endogenously synthesize only 11 AAs, known as non-essential AAs (NEAAs)²⁶, and have to obtain the remaining nine AAs, known as essential AAs (EAAs), from the diet²⁷ or from microbes²⁸⁻³¹. However, the endogenous synthesis of NEAAs might not be sufficient for the proliferation of cancer cells, since reduced exogenous supply of NEAAs such as glutamine can impair the survival or tumorigenic potential of malignant cells^{17,20,32-37}. In fact, reduced dietary content of NEAAs even limits protein synthesis and growth of normal tissues^{38,39}. Importantly, recent metabolic profiling experiments have demonstrated that cancer cells obtain EAAs and some NEAAs from external sources for protein synthesis^{18,40,41}.”

2) In Discussion, we further emphasize this point with the following sentences (Line 371–377): “Previous studies have demonstrated that cancer cells may employ multiple strategies to acquire AAs^{17,32}, such as the endogenous synthesis of NEAAs^{18,37,40,41,82}, up-regulating AA transporting^{20,41,74}, or through micropinocytosis⁸³⁻⁸⁵. In this study, we investigated how the management of AA usage confers advantages on cancer cells during tumor progression. Since the use of all 20 AAs in human proteomes is constrained by their synthetic costs in the living organisms, our ECPA concept effectively reflects how cancer cells optimize gene expression profiles for AA usage adaptation. We revealed a common principle governing cancer evolution: cancer cells evolve to use AAs more economically by down-regulating the genes rich in costly AAs.”

3) Furthermore, we emphasized that ECPA reflects how efficiently cancer cells manage the usage of AAs based on their price-tag in Discussion with the following sentences (Line 390–403): “Human intracellular AAs consist of EAAs ultimately taken from the environment and NEAAs synthesized endogenously in human cells. Our ECPA_{cell} analysis based on the Y20 (or B20) cost metric well reflects how cancer cells manage the use of 20 AAs during proliferation. We observed similar overall patterns when we focused on only the 11 NEAAs (H11), presumably due to the conservation of biosynthetic costs for these NEAAs in humans and yeast (or bacteria). Notably, the results based on the H11 metric are also consistent with the hypothesis that cancer cells reduce the overall energy costs in synthesizing the 11 NEAAs, which would confer advantages to cancer cells that allow them to proliferate more rapidly. Nevertheless, recent studies demonstrate that cancer cells need to uptake EAAs and a large amount of NEAAs from the

environment during proliferation, probably through a cancer-type-specific manner^{20,32,74}. Tumorigenesis is an evolutionary process and the populations of cancerous cells evolve like populations of organisms¹⁻⁶. Noteworthy, the selective pressure acting at the organism level is different from that experienced by the cancer cells. While human normal cells are required to maintain AA homeostasis for normal biological function, cancer cells with lower ECPA_{cell} would gain the selective advantage over the other cells in tumor microenvironment as free AAs of lower biosynthetic costs are more abundant in that environment.”

4) In this revision, we inserted one sentence to explain why we observed stronger anti-correlation between the ECPA_{gene} and the gene expression levels in tumors than in normal tissues (Line 172–173): “These results suggest that cancer cells might more efficiently manage the available AAs in the microenvironments for protein synthesis.”

5) As mentioned above, in the revised manuscript, we further clarify that ECPA in human cells do not reflect the actual energy human cells invest in synthesizing the AAs (Line 72–77 of Page 4) with the following sentences: “Since the EAAs and most NEAAs in human cells are ultimately taken from the autotrophs, neither ECPA_{gene} nor ECPA_{cell} measures the actual energy human cells invest to synthesize the AAs endogenously. Rather, these parameters can be treated as the average “price-tag” for the AAs in a protein or the proteomes, respectively. As the relative abundance of AAs is anti-correlated with the biosynthetic cost in the living organisms, the ECPA metrics inform how efficiently human cells manage the given AA pool for protein synthesis.”

As for the critique that “the magnitude of these effects is relatively small, bringing their biological or functional relevance under question”, we respectfully disagree with this reviewer. As pointed out by Reviewer #1, our results were based on solid statistical analyses. In fact, the effect size (can be viewed as a kind of standardized magnitude of effects) for differences in ECPA is quite large in our analyses because the variation in ECPA is also very small. Furthermore, for many types of cancers, we found ECPA_{cell} values are significantly lower in the tumor than normal tissues, and the ECPA_{cell} values are significantly associated with tumor progression and the survival time of patients. Importantly, tumorigenesis is an evolutionary process and the populations of cancerous cells evolve like populations of organisms (Cairns, 1975; Korolev et al., 2014; McGranahan and Swanton, 2017; Merlo et al., 2006; Nowell, 1976; Wu et al., 2016). From the view of population genetics, a slight change in the selective coefficient can lead to profound impact on the fitness after many generations (Charlesworth, 2009; Kimura, 1964, 1983). As shown in our simulation results (Line 273–279), even a slight selective advantage conferred by the reduced ECPA_{cell} values would eventually significantly increase the fraction of the cancer cells that have reduced ECPA_{cell} values in the cancer cell population (Fig. 5b-d).

6. Significant limitations in interpretation of data in the context of contemporary thinking in cancer biology and clinical management of disease lessen the potential interest and utility of this manuscript to the cancer metabolism field, and cancer biology more broadly.

Response: We are not quite sure about what the reviewer wants us to improve here. Besides the efforts as above described, in this revision we also conducted new analyses based on your comments and suggestions in the below sections. Briefly, some of our efforts (in Line 198–211) can be summarized as follows:

- 1) We analyzed ECPA_{cell} of tumor samples from different subtypes of breast carcinoma (Ciriello et al., 2015) which has the largest number of samples in the TCGA data. Compared to the normal samples, all tumor subtypes have significantly lower ECPA_{cell} (Supplementary Fig. 12). These results suggest that the reduced ECPA_{cell} in cancer cells is robust with respect to tumor subtypes.
- 2) We performed the ECPA_{cell} analysis on previously published single-cell RNA-Seq data of melanoma (Tirosch et al., 2016) and ovarian carcinoma cells (Schelker et al., 2017) to assess the influence of the heterogeneous cellular composition in the cancer samples. For both cancer types, the malignant cancer cells have significantly lower ECPA_{cell} values than the immune or stromal cells (Supplementary Fig. 13), suggesting that the reduced ECPA_{cell} in tumors are mainly shaped by the malignant cells rather than the immune and stromal cells in the tumor microenvironment.
- 3) We considered the somatic mutations that might change the protein sequences when calculating the ECPA_{cell} values in each tumor sample. Since the number of AA changes caused by somatic

mutations in a cancer sample is small (20~100)(Cancer Genome Atlas Research et al., 2013), we found such AA changes have negligible effects on the observed difference in $ECPA_{cell}$ values between the normal and cancer samples (Supplementary Fig. 14).

Is the work convincing, and if not, what further evidence would be required to strengthen the conclusions?

7. The manuscript is based on an overly simplistic assumption that *de novo* amino acid synthesis is the predominant source of amino acids for protein synthesis in cancer cells. This ignores a huge and rapidly evolving body of evidence about both the source and use of amino acid in cancer cells. Significant sources of amino acids in cancer cell include amino acid import, recycling via autophagy, and protein degradation by the proteasome. Similarly, the metabolic fate of amino acids extends beyond protein synthesis to include biosynthesis of nucleotides, fatty acids, reductive intermediates, etc. In other words, what is the relative importance/contribution of amino acid biosynthesis to cancer cell proliferation?

Response: These comments seem to be based on the assumption that we assumed *de novo* biosynthesis is the primary source of AAs used for protein synthesis. Obviously, human cells cannot synthesize essential AAs (EAAs) and rely on the environment or protein turnover for the supply of both EAAs and most non-essential AAs (NEAAs), as we acknowledged in Introduction and Discussion of our initial submission. In this revision, we made substantial changes in the revised manuscript to emphasize that our $ECPA$ metric in human cells reflects the management of the utilization of the 20 AAs in the proteome. In other words, $ECPA_{gene}$ or $ECPA_{cell}$ can be treated as the average “price-tag” for the AAs in a protein or in the proteomes, respectively, and neither parameter measures the actual energy human cells invest to synthesize all the AAs, because the EAAs and most NEAAs in human cells are ultimately taken from the autotrophs. Please refer to our response to you at comment # 5 for details.

8. Overall, the biological interpretations and conclusions are not supported by the experimental data and are highly speculative. The concept that amino acid biosynthetic cost plays a major role in tumour biology and evolution needs to be tested experimentally (e.g. measure changes in the relative contribution of individual AA in various metabolic pathways like protein, lipid, nucleotide synthesis, or demonstrate functional changes in tumour or cell behavior following inhibition or manipulation of *de novo* amino acid biosynthesis).

Response: Again, these comments are based on the misunderstanding of $ECPA$ metric. We apologize for not making the concept of $ECPA$ clear enough in our initial submission. Please refer to our response to your comment #5 and comment #7 for details.

9. Have the authors excluded the possibility that the observed small changes in $ECPA$ are a proxy for some other effect such as proliferation?

Response: Thanks for raising this concern. This is exactly the same concern Reviewer #2 raised. To address this issue, in this revision, we compiled a list of 530 proliferation-related genes from a recent study (Waldman et al., 2013). These proliferation-related genes were identified because their expression levels showed significant positive association with the growth rate of the cancer cells among NCI-60 cancer cell lines. We found these proliferation-related genes are up-regulated in cancer cells compared to normal tissue (Supplementary Fig. 28a). We found the proliferation-related genes do have lower $ECPA_{gene}$ than the remaining genes (Supplementary Fig. 28b). However, the association between $ECPA_{cell}$ and the cancer progression in the pan-cancer analysis remains intact when we excluded the proliferation-related genes from the analysis (Supplementary Fig. 29). The reduction of $ECPA_{cell}$ in experimental evolution of xenograft tumors is not affected by proliferation-related genes either (Supplementary Fig. 31). Furthermore, the $ECPA_{cell}$ value calculated with only the proliferation-related genes is not significantly associated with the patient survival time (Supplementary Fig. 30). These results suggest that the whole patterns we observed are not likely caused by the changes in proliferation-related genes alone.

We report the updated results in Line 308–316 of Page 12 as follows:

“The expression levels of proliferation-related genes⁷⁵ are increased in tumors compared to the matched normal samples (Supplementary Fig. 28a). Although the proliferation-related genes have lower ECPA_{gene} than the remaining genes (Supplementary Fig. 28b), the results of pan-cancer analyses are only slightly affected after excluding these genes (Supplementary Fig. 29). Also, the ECPA_{cell} values calculated with only the proliferation-related genes do not consistently show association with the patient survival time (Supplementary Fig. 30). Furthermore, the reduction of ECPA_{cell} during experimental evolution of xenograft tumors still holds when the proliferation-related genes were excluded (Supplementary Fig. 31). These results suggest the association between ECPA_{cell} and the cancer progression is unlikely caused by the changes in proliferation-related genes alone.”

10. Further evidence supporting the conclusions would also need to include consideration of tumour subtypes and comparison of the putative prognostic value of ECPA against multiple gene expression panels.

Response: Thank you for these constructive comments. Following this reviewer’s suggestion, we examine the influence of tumor subtypes on the ECPA_{cell} of cancer cells. Briefly, we retrieved the PAM50 molecular subtypes of breast invasive carcinoma (BRCA) from the original TCGA study (Ciriello et al., 2015). Compared to normal samples, we found tumor samples of all subtypes have significantly lower ECPA_{cell}, which was consistent with the result obtained when pooling all subtypes together. Therefore, the differences in ECPA_{cell} between tumor and normal samples are not subtype-specific. We reported the new results in Line 199–202 of this revised manuscript.

The main purpose of this study is to deepen our understanding of the general pattern how economical usage of AAs would affect the evolution of cancer cells. We are glad to find that ECPA outperforms most single gene-based markers with respect to the prognostic power. However, our results do not necessarily suggest ECPA is better than pre-existing gene expression-based markers for a specific type of cancer. After all, as Reviewer # 3 concluded: “it is good to have a novel and independent marker for prognosis.”

11. Altered gene expression is only one aspect reflecting underlying functional and genetic changes in cancer cells. These cells also carry a significant mutational load. What is the potential contribution of mutations (particularly those that affect protein stability and turnover through altered post-translational modifications) on energy cost of biosynthesis?

Response: Thanks for the suggestions. While there is considerable mutation load in cancers (Cancer Genome Atlas Research et al., 2013), very few of them are shared between different cases (Wu et al., 2016). Therefore, somatic mutations might be unlikely to explain the differences in ECPA_{cell} between tumor and normal samples. To evaluate whether the reduction in ECPA_{cell} values in tumors are mainly caused by DNA mutations that change expensive AAs into cheap AAs, we incorporated the somatic mutations that change the protein sequences and calculated the ECPA_{cell} values in each tumor sample. Since the number of AA changes caused by somatic mutations in a cancer sample is small (20~100)(Cancer Genome Atlas Research et al., 2013); we found such AA changes have negligible effects on the observed difference in ECPA_{cell} values between the normal and cancer samples (Supplementary Fig. 14). We reported the new results in Line 208–211 of this revised manuscript.

12. P9: Regarding the statement that “ECPA_{cell} is an essential cellular property in cancer evolution” - Gene expression data used in these calculations is largely collected at the tissue/tumour level and does not provide single cell resolution. In this context, ECPA_{cell} is a somewhat misleading term, as energy cost has not been calculated at the single cell level. This also ignores the contribution of stromal and immune cell populations to gene expression signatures. These can be a significant component of tumours (e.g. in pancreatic cancer). Single cell transcriptomes are now available for a number of cancer

types and analysis of these data in the context of ECPA_{cell} would be much more informative, and relevant to tumour biology and evolution.

Response: Thanks for this suggestion. In our previous submission, we mainly used the mRNA-Seq data generated from the bulk sequencing of cell populations as they reflected the general trends of cell populations and are more accurate than the single-cell transcriptome sequencing (Svensson et al., 2017). The major concern of the reviewer is that our analyses were based on bulk sequencing data of cell populations, which are heterogeneous as immune and stromal cells are also present in tumor microenvironment (Hanahan and Weinberg, 2011) except for malignant cells. To address this concern, in this revision we analyzed the previously published single-cell RNA-Seq data of melanoma (Tirosh et al., 2016) and ovarian carcinoma cells (Schelker et al., 2017). For both cancer types, the malignant cancer cells have significantly lower ECPA_{cell} values than the immune or stromal cells (Supplementary Fig. 13). These results suggest that the reduced ECPA_{cell} in tumors are mainly shaped by the malignant cells rather than the immune and stromal cells in the tumor microenvironment. We present the new results in Line 202–208 as follows:

“Besides malignant cells, immune and stromal cells are also present in tumor microenvironment¹⁰. To assess the influence of the heterogeneous cellular composition in the cancer samples, we performed the ECPA_{cell} analysis on previously published single-cell RNA-Seq data of melanoma⁶⁷ and ovarian carcinoma cells⁶⁸. For both cancer types, the malignant cancer cells have significantly lower ECPA_{cell} values than the immune or stromal cells (Supplementary Fig. 13), suggesting that the reduced ECPA_{cell} in tumors are mainly shaped by the malignant cells rather than the immune and stromal cells in the tumor microenvironment.”

13. P8: Please provide evidence for limited availability of AAs in the tumour microenvironment?

Response: Thank you for reminding. The limited AA supply in tumor microenvironment has been reported previously and widely accepted (Lindqvist et al., 2018; Pavlova and Thompson, 2016; Vander Heiden and DeBerardinis, 2017). We present the relevant citations in Line 257–259 with the sentence: “Cancer cells with alterations that reduce ECPA_{cell} (*i.e.*, more economical use of AAs in protein synthesis) have advantages over other cancer cells and proliferate more rapidly given limited AAs in the microenvironment^{12,17,73}.”

14. P10: “strongly suggest that ECPA_{cell} is selected for reduction during tumour evolution... shaped the evolutionary roadmap”. This claim is somewhat hyperbolic given the very small effect measured.

Response: Thanks for raising this concern. In this revision we have rephrased the relevant sentences as follows: “This *in vivo* experimental study supports the notion that ECPA_{cell} is selected for reduction during tumor evolution” (Line 271–272), and “Collectively, our experimental evolution and simulations both suggest that reduced ECPA_{cell} may be an important feature of tumor cells during cancer progression.” (Line 279–280 Page 11).

On a more subjective note, do you feel that the paper will influence thinking in the field?

15. As the paper doesn't reflect or incorporate several aspects of contemporary thinking in cancer cell metabolism, cellular heterogeneity, tumour subtype, etc., its potential to influence thinking in the field is limited.

Response: We are grateful for your critiques and suggestions, which are valuable and help improve this study significantly. In this revision, we have further clarified the concept of ECPA as described in the responses to your comment #4, #5, and #7. We also followed your suggestions by performing ECPA analyses on different tumor subtypes and analyzing single-cell RNA-seq data to control for cellular heterogeneity. Please refer to our responses to your comment #10 and #12 for details. We hope the revised manuscript is satisfactory to you.

Further questions and concerns about the paper.

16. Throughout the paper, gene expression measurements used were collected at the tissue (i.e. tumour) level and represent the “average” expression of a gene across all cells present in the sample (including stroma), but interpretation and discussion often refers to single cell level effects. For example, on page 4 “Based on ECPA_{cell} and the overall gene expression profile of a cell...”. This is a significant error that must be addressed.

Response: Thank you for the suggestion. To address your concern, we rephrased the relevant sentence as follows: “Based on ECPA_{gene} and the overall gene expression profile of a sample, we calculate ECPA_{cell}, ...” (Line 70–71 in the revised manuscript). We also emphasize this point by inserting the sentence “Since the TCGA mRNA-Seq expression data were quantified at the tissue level, the ECPA_{cell} value represents the “average” virtual cost of proteinic AAs across all cells present in that sample.” in Line 191–192 of this revised manuscript.

Following your suggestion, we also evaluated the influence of stroma and immune cells in the tumor microenvironment with previously published single-cell RNA-Seq data. We present the new results of single-cell analysis as follows (in Line 202–208):

“Besides malignant cells, immune and stromal cells are also present in tumor microenvironment¹⁰. To assess the influence of the heterogeneous cellular composition in the cancer samples, we performed the ECPA_{cell} analysis on previously published single cell RNA-Seq data of melanoma⁶⁷ and ovarian carcinoma cells⁶⁸. For both cancer types, the malignant cancer cells have significantly lower ECPA_{cell} values than the immune or stromal cells (Supplementary Fig. 13), suggesting that the reduced ECPA_{cell} in tumors are mainly shaped by the malignant cells rather than the immune and stromal cells in the tumor microenvironment.”

17. P8: It is important to consider the critical role of heterogeneity in gene expression profiles between individual patients. Equally, and using the same logic, heterogeneity of gene expression between individual cells within a tumour is a very important consideration that has not been measured in this analysis (see comments above)

Response: Thanks for this suggestion. We have addressed the issue of heterogeneity in response to your comment #10 and #12.

First, we evaluated the influence of heterogeneity between individual patients by analyzing different subtypes of breast cancer separately. We found the reduced ECPA_{cell} is not subtype-specific. We present the new results in Line 197–201 and the relevant sentences are reproduced as follows: “Within a cancer type, the gene expression profiles of different patients are highly heterogeneous. Hence, we analyzed ECPA_{cell} of tumor samples from different subtypes of breast carcinoma⁶⁶ which has the largest number of samples in the TCGA data. Compared to the normal samples, all tumor subtypes have significantly lower ECPA_{cell} (Supplementary Fig. 12), suggesting that the reduced ECPA_{cell} in cancer cells is robust in respect to tumor subtypes.”

Second, we also analyzed the previously published single-cell RNA-Seq data of melanoma (Tirosh et al., 2016) and ovarian carcinoma cells (Schelker et al., 2017). For both cancer types, the cancer cells have significantly lower ECPA_{cell} values than the immune or stromal cells (Supplementary Fig. 13). These results suggest that the reduced ECPA_{cell} in tumors is mainly shaped by the malignant cells rather than the immune and stromal cells in the tumor microenvironment. We present the new results in Line 202–208 as follows:

“Besides malignant cells, immune and stromal cells are also present in tumor microenvironment¹⁰. To assess the influence of the heterogeneous cellular composition in the cancer samples, we performed the ECPA_{cell} analysis on previously published single cell RNA-Seq data of melanoma⁶⁷ and ovarian carcinoma cells⁶⁸. For both cancer types, the malignant cancer cells have significantly lower ECPA_{cell} values than the immune or stromal cells (Supplementary Fig. 13), suggesting that the reduced ECPA_{cell} in tumors are mainly shaped by the malignant cells rather than the immune and stromal cells in the tumor microenvironment.”

18. P9, re: “superiority of ECPA_{cell} over individual genes as a potential prognostic marker”. The data presented do not support this claim as the analysis ignores the important distinction of cancer subtypes based on cell and genetic markers, which are an important part of clinical decision making on treatment

and prognosis). Further, as single gene markers are rarely used, the clinical utility of this claim is questionable in contemporary management of cancer.

Response: Thanks for raising this concern. In the revised manuscript, we have rephrased the offending sentence into “These results also highlight the feasibility of ECPA_{cell} as potential prognostic markers for patient stratification.” (Line 252–253).

As we mentioned above, in this revision, we also evaluated the influence of heterogeneity between individual patients by analyzing different subtypes of breast cancer separately. We found the reduced ECPA_{cell} is not subtype-specific. We present the new results in Line 198-202 and the relevant sentences are reproduced as follows: “Within a cancer type, the gene expression profiles of different patients are highly heterogeneous. Hence, we analyzed ECPA_{cell} of tumor samples from different subtypes of breast carcinoma⁶⁶ which has the largest number of samples in the TCGA data. Compared to the normal samples, all tumor subtypes have significantly lower ECPA_{cell} (Supplementary Fig. 12), suggesting that the reduced ECPA_{cell} in cancer cells is robust in respect to tumor subtypes.”

19. P11: re low ECPA_{cell} as a marker of resistance to immunotherapies. How does this, measure compare with PD1/PDL1 expression?

Response: Thanks for raising this concern. Recent studies suggest the response to anti-PD-1/PD-L1 is more likely to be observed in patients with high expression level of PD-L1 in tumors for many cancer types. However, there is no significance differences in the mRNA levels of gene *PD-1* (*PDCDI*) or *PD-L1* (*CD274*) between the responding group and the non-responding group (Wilcoxon rank-sum tests; $P = 0.82, 0.53$ for *PD-1* and *PD-L1*, respectively) in this cohort. Similar observation was also made in a recent study (Riaz et al., 2017). We further found the expression of *PD-1* and *PD-L1* was also dysregulated in tumors compared to normal samples (Supplementary Fig. 34a and 35a). However, the expression level of neither gene showed consistent associations with the pathological stage of tumors or patient survival time (Supplementary Fig. 34 and 35). This comparison suggests that the expression of *PD-1* or *PD-L1* is less optimal than ECPA in tumor prognosis or predicting response to immunotherapy. This has been reported in Line 336–340 and Line 345–347 as follows:

“The expression levels *PD-1* (*PDCDI*) or *PD-L1* (*CD274*) are associated with the response to checkpoint blockade therapy^{77,78}. Although *PD-1* and *PD-L1* are usually dysregulated in tumors compared to normal samples (Supplementary Fig. 34a and 35a), the expression level of neither gene showed consistent associations with the pathological stage of tumors or patient survival time (Supplementary Fig. 34 and 35). ... By contrast, we didn’t find significant differences in expression levels for *PD-1* ($P = 0.82$) or *PD-L1* ($P = 0.53$) between patients of the responding group and the non-responding group, which is consistent with a recent study⁸⁰.”

20. P12: The claim about the predictive power of ECPA_{cell} was not actually tested in a predictive context, it is based on a single cohort and should be tested against other models and cohorts.

Response: Thanks for raising this concern. Focusing on the 17 cancer types with sufficient samples and events (Methods and Supplementary Fig. 10), we found that patients with lower ECPA_{cell} show significantly worse survival probability compared to those with higher ECPA_{cell} in 9 cancer types; and we did not find a significantly reversed pattern in any cancer type (split by the median ECPA value, log-rank test, Fig. 4c, 4d). Further, a lower ECPA_{cell} was significantly associated with poor survival using a univariate Cox proportional hazards model in the nine cancer types. Collectively, in 11 of the 17 cancer types surveyed, lower ECPA_{cell} showed a significant correlation with poorer patient prognosis by either log-rank test or Cox model (see additional cancer types in Supplementary Fig. 10). To confirm the statistical significance of the observed pattern, we performed permutation tests on cancer samples and found that the number of cancer types with consistent survival correlation was much higher than the random expectation (at most 5 in permutations, $P < 2 \times 10^{-4}$, Supplementary Fig. 12a). Importantly, in six cancer types, the association with ECPA_{cell} remained significant even when the pathologic tumor stage and patient age were considered in the multivariate analysis (Fig. 4c), which suggests that ECPA_{cell} provides additional prognostic power over clinical variables. For comparison, we stratified patients by the expression level of individual genes and tested their associations with the pathological stage or

patient survival time. Among 18,919 genes surveyed, only one gene (*LOX*) showed a comparable, consistent association with both pathological stage and survival analysis, and the probability of a gene with similar prognostic power across multiple cancers was 2.1×10^{-4} (Supplementary Fig. 12b). Indeed, when examining a set of known cancer therapeutic targets or biomarker genes (Van Allen et al., 2014), none of them showed such as consistent a prognostic pattern as $ECPA_{cell}$ (Supplementary Fig. 12c). Since we consistently found $ECPA_{cell}$ is reduced in tumor samples from many different cancer types, these results cross-validate the predictive power of $ECPA_{cell}$.

21. P12: The claim that “the use of all 20 AAs in human proteomes is constrained by their synthetic costs” is rather broad and must be supported by cited references.

Response: Thank you for raising this point. In this revised manuscript, we have rephrased the sentence into “Since the use of all 20 AAs in human proteomes is constrained by their synthetic cost in the living organisms” in the revised manuscript. In fact, we have spent a lot of efforts on this point in both the original and the revised submission.

In the first part of the manuscript, we analyzed more than 10,000 genomes and found the anti-correlation between the biosynthetic energy cost and utilization of amino acid in all the domains of life (Line 82–108). Our results support the previous hypothesis that in the autotrophs (bacteria, yeast, and plants which can synthesize all 20 proteinogenic AAs), the anticorrelation between the biosynthetic cost and usage of AAs is driven by natural selection for bioenergetic efficiency (Akashi and Gojobori, 2002). Interestingly, although animals can synthesize only 11 NEAAs (Payne and Loomis, 2006), we also observed significant C-U anticorrelations for all the 20 AAs in humans and other animals when the cost of AA biosynthesis in bacteria (B20) or yeast (Y20) is employed. Since the biosynthetic pathway of NEAAs might be different in humans and other animals compared to yeast or bacteria (Lehninger et al., 2008), we calculated the biosynthetic cost for each NEAA in humans (H11) following previous studies in bacteria (Akashi and Gojobori, 2002) or yeast (Raiford et al., 2008; Wagner, 2005), while taking into account the differences. With the H11 cost index, we also detected significant C-U anti-correlation of AAs in humans and other animals. Our results are in line with the notion that EAAs and most NEAAs in animal cells are ultimately taken from the autotrophs in which the bioavailability of an AA is constrained by its biosynthetic cost (Heizer et al., 2011; Krick et al., 2014; Swire, 2007).

In this revision, we highlight the difference between autotrophs and heterotrophs with the following sentences (Line 117-124): “Our results support the notion that in autotrophs (bacteria, yeast, and plants), which can synthesize all 20 proteinogenic AAs, the C-U anticorrelation of AAs is driven by natural selection for bioenergetic efficiency⁴². In line with previous discoveries^{45,46,52}, we also observed significant C-U anticorrelations for all the 20 AAs in humans and other animals even though animals can synthesize only 11 NEAAs²⁶. Since EAAs and most NEAAs in animal cells are ultimately taken from the autotrophs in which the bioavailability of an AA is constrained by its biosynthetic cost^{45,46,52}, in Fig. 2a, we illustrate how the biosynthetic energy cost underlies the usage of AAs in human proteomes.”

We also conducted simulations to show that the mixtures of AAs from the two sources: (NEAAs endogenously synthesized in animal cells and all the 20 AAs ultimately taken from autotrophs) yield significantly negative correlations between the overall abundance and cost of all 20 AAs in autotrophs (Supplementary Fig. 6). Moreover, we also compiled previous the experimental data show that the biosynthetic cost is significantly anticorrelated with the abundance of free AAs in human and animal tissues (Fig. 2b). Altogether, we conclude that the biosynthetic costs of all 20 AAs constrain the relative abundances of free AAs in human cells, which further shapes AA usage in the proteomes (Fig. 2a).

22. P12: re claim that cancer cells evolve to use AAs more efficiently was not actually tested as measurements were made at the cellular level.

Response: If our understanding is correct, this reviewer expects us to present evidence to show that the absolute rate of AA consumption is enhanced in cancer cells. If that is true, the reviewer might misunderstand our thesis. The $ECPA$ metric is similar to the body composition index that reflects the relative but not the absolute usage of AAs in the cells. Our $ECPA$ analyses suggest that cancer cells

prefer to avoid using the AAs that are expensive to synthesis in the living organisms (not necessarily endogenously synthesized in human cells) given the available AA pools in the cancer microenvironment. We observed a clear decreasing trend of ECPA_{cell} in a temporal order of the eight xenograft tumors (Fig. 5a), and this *in vivo* experimental study supports the notion that human cancer cells optimized their expression profiles during evolution to adjust the usage of AAs during tumor evolution.

23. P13: Individual genes are increasingly rarely used as prognostic markers, so comparing the power of ECPA_{cell} against these is somewhat outdated. Further, many tumour types are increasingly being parsed into subtypes based on gene expression or mutational signatures for diagnostic, prognostic and therapeutic use. The performance of ECPA_{cell} as a prognostic marker needs to be evaluated in this context.

Response: As we mentioned above, the main purpose of this study is to deepen our understanding of the general pattern how economical usage of AAs would affect the evolution of cancer cells. We are glad to find that ECPA has better prognostic power than most single gene-based markers. However, our results do not necessarily suggest ECPA is better than pre-existing gene expression-based markers for a specific type of cancer. After all, as Reviewer # 3 concluded: "it is good to have a novel and independent marker for prognosis."

In this revision, we also evaluated the influence of heterogeneity between individual patients by analyzing different subtypes of breast cancer separately. We found the reduced ECPA_{cell} is not subtype-specific. We present the new results in Line 199–202 and the relevant sentences are reproduced as follows: "Hence, we analyzed ECPA_{cell} of tumor samples from different subtypes of breast carcinoma⁶⁶ which has the largest number of samples in the TCGA data. Compared to the normal samples, all tumor subtypes have significantly lower ECPA_{cell} (Supplementary Fig. 12), suggesting that the reduced ECPA_{cell} in cancer cells is robust with respect to tumor subtypes."

24. P13/14: How does ECPA_{cell} factor in the cost of AA acquisition from the environment, which represents a significant source of AAs in tumour cells? In this context, wouldn't there be a bias in tumour cell requirements for biosynthesis of NEAA?

Response: This concern seems to be based on the misunderstanding of the concept of ECPA. This reviewer is respectfully directed to our response to your comment #10 for details. Moreover, in our previous submission, we summarized previous studies showing that cancer cells would employ multiple strategies to acquire AAs with the sentences (Line 371–377): "Previous studies have demonstrated that cancer cells may employ multiple strategies to acquire AAs^{17,32}, such as the endogenous synthesis of NEAAs^{18,37,40,41,82}, up-regulating AA transporting^{20,41,74}, or through micropinocytosis⁸³⁻⁸⁵. In this study, we investigated how the management of AA usage confers advantages on cancer cells during tumor progression. Since the use of all 20 AAs in human proteomes is constrained by their synthetic costs in the living organisms, our ECPA concept effectively reflects how cancer cells optimize gene expression profiles for AA usage adaptation. We revealed a common principle governing cancer evolution: cancer cells evolve to use AAs more economically by down-regulating the genes rich in costly AAs."

25. P13: How do ECPA_{cell} values compare with protein synthesis? This can be measured experimentally in the context of inhibition of protein synthesis.

Response: Thanks for raising this concern. It is well established that during mRNA translation, roughly 4 ~P is used for each amino acid to be added to the elongating peptide (Wagner, 2005). Our ECPA_{cell} reflects how human cells efficiently use the AAs based on their bioavailability, which is constrained by their synthetic cost in the living organisms. Since human cancer cells obtain most AAs from the environment rather than by endogenous synthesis, we don't think we should directly compare ECPA_{cell} with the number of ~P the cells consumed for the mRNA translation process. Therefore, the suggested experiment is not relevant to our study. The most appropriate data for ECPA calculation should be the biosynthesis rates of all the proteins in cells. Ribosome profiling is an effective way to measure genome-wide protein translation rate (Ingolia, 2014, 2016; Ingolia et al., 2009). We verified our observation that

ECPA_{cell} is significantly lower in tumor samples than in matched normal samples with the ribosome profiling data from a recent study (Loayza-Puch et al., 2016). This result could be found in Line 347–335 of Page 13–14 in our original manuscript:

“To further validate our analysis, we retrieved the ribosome profiling data of normal ($n=6$) and tumor ($n=10$) samples of human kidney tissue ⁷⁹ and calculated ECPA_{cell} with the ribosome-protected fragments (Supplementary Methods). Consistent with the observation using TCGA mRNA-Seq data, we found that ECPA_{cell} in the tumor samples is significantly lower than that in the normal samples whenever we used the Y20, B20 or H11 metric (Supplemental Fig.S21).”

26. Fig1F regression does not look correct. There are 2 clear clusters in the data?

Response: Thanks for pointing this out. Actually, there are some outliers on the right side of the plot rather than another cluster.

27. Fig3C: need to see individual correlations, not just bar plots.

Response: Thanks for the suggestion. Following your suggestion, we provided the individual correlations in Supplementary Fig. 8 of this revised manuscript.

28. Fig 3D: Scales of graphs are misleading. Magnitude of effect is very small here.

Response: Here we would like to highlight the anti-correlation between ECPA_{gene} and the expression levels of genes (binned). As we described above (in response to comment # 5), tumorigenesis is an evolutionary process and the populations of cancerous cells evolve like populations of organisms (Cairns, 1975; Korolev et al., 2014; McGranahan and Swanton, 2017; Merlo et al., 2006; Nowell, 1976; Wu et al., 2016). From the view of population genetics, a slight change in the selective coefficient can lead to profound impact on the fitness after many generations. As shown in our simulation results (Line 273–279), even a slight selective advantage conferred by the reduced ECPA_{cell} values would eventually significantly increase the fraction of the cancer cells that have reduced ECPA_{cell} values in the cancer cell population (Charlesworth, 2009; Kimura, 1964, 1983). Therefore, we respectfully disagree with this reviewer on the critique “Magnitude of effect is very small here.”

29. Fig4A graph scale condensed, magnitude of effect is very small. Need to split cancers by subtype.

Response: Thanks for this suggestion. As mentioned above, we evaluated the influence of heterogeneity between individual patients by analyzing different subtypes of breast cancer separately. We found the reduced ECPA_{cell} is not subtype-specific. We present the new results in Line 199–202 and the relevant sentences are reproduced as follows: “Hence, we analyzed ECPA_{cell} of tumor samples from different subtypes of breast carcinoma ⁶⁶ which has the largest number of samples in the TCGA data. Compared to the normal samples, all tumor subtypes have significantly lower ECPA_{cell} (Supplementary Fig. 12), suggesting that the reduced ECPA_{cell} in cancer cells is robust with respect to tumor subtypes.”

30. Fig5a: regression fits are not convincing. Appears to be an initial increase in ECPA_{cell}, followed by a subsequent decrease. Again, magnitude of effect is very small.

Response: Thank you for the comments. We agree with this reviewer that the ECPA_{cell} values fluctuate along with the stage of the xenograft tumor during the experimental evolution. However, the general trend revealed by the regression fit suggests that ECPA_{cell} decreases during tumor evolution. Importantly, our analysis is conservative since we excluded the initial stage (MCF10A-hRas, the black point in Fig. 5a) and the metastasis tumors (the blue points in Fig. 5a) in the regression analysis. Including points for those stages in the analysis would make the trend of ECPA_{cell} decreasing more statistically significant.

31. Fig 7a: again, magnitude of effect is very small.

Response: Thanks for pointing this out. As shown in our simulation results (Line 273–279), even a slight selective advantage conferred by the reduced ECPA_{cell} values would eventually significantly increase the fraction of the cancer cells that have reduced ECPA_{cell} values in the cancer cell population (Charlesworth, 2009; Kimura, 1964, 1983). We also further confirmed the significance of the observed differences with permutation analysis (Line 349–354):

“To further confirm that the observed significant pattern is due to the biosynthetic costs of different AAs, we randomly permuted the biosynthetic energy costs of AAs 1,000 times, repeated the above analysis between the two response groups, and visualized the obtained *P*-values and ECPA_{cell} differences [$\log_2(\text{responding}/\text{non-responding})$] using a volcano plot (Fig. 7b). We found that the ECPA_{cell} difference using the real biosynthetic energy costs of AAs was significantly larger than that using the shuffled energy costs ($P = 0.012$, Fig. 7b).”

References

- Akashi, H., and Gojobori, T. (2002). Metabolic efficiency and amino acid composition in the proteomes of *Escherichia coli* and *Bacillus subtilis*. *Proceedings of the National Academy of Sciences of the United States of America* 99, 3695-3700.
- Cairns, J. (1975). Mutation selection and the natural history of cancer. *Nature* 255, 197-200.
- Cancer Genome Atlas Research, N., Weinstein, J.N., Collisson, E.A., Mills, G.B., Shaw, K.R., Ozenberger, B.A., Ellrott, K., Shmulevich, I., Sander, C., and Stuart, J.M. (2013). The Cancer Genome Atlas Pan-Cancer analysis project. *Nature genetics* 45, 1113-1120.
- Charlesworth, B. (2009). Fundamental concepts in genetics: effective population size and patterns of molecular evolution and variation. *Nature reviews Genetics* 10, 195-205.
- Ciriello, G., Gatza, M.L., Beck, A.H., Wilkerson, M.D., Rhie, S.K., Pastore, A., Zhang, H., McLellan, M., Yau, C., Kandoth, C., *et al.* (2015). Comprehensive Molecular Portraits of Invasive Lobular Breast Cancer. *Cell* 163, 506-519.
- Hanahan, D., and Weinberg, R.A. (2011). Hallmarks of cancer: the next generation. *Cell* 144, 646-674.
- Heizer, E.M., Raymer, M.L., and Krane, D.E. (2011). Amino Acid Biosynthetic Cost and Protein Conservation. *Journal of Molecular Evolution* 72, 466-473.
- Hosios, Aaron M., Hecht, Vivian C., Danai, Laura V., Johnson, Marc O., Rathmell, Jeffrey C., Steinhauser, Matthew L., Manalis, Scott R., and Vander Heiden, Matthew G. (2016). Amino Acids Rather than Glucose Account for the Majority of Cell Mass in Proliferating Mammalian Cells. *Developmental Cell* 36, 540-549.
- Ingolia, N.T. (2014). Ribosome profiling: new views of translation, from single codons to genome scale. *Nature reviews Genetics* 15, 205-213.
- Ingolia, N.T. (2016). Ribosome Footprint Profiling of Translation throughout the Genome. *Cell* 165, 22-33.
- Ingolia, N.T., Ghaemmaghami, S., Newman, J.R., and Weissman, J.S. (2009). Genome-wide analysis in vivo of translation with nucleotide resolution using ribosome profiling. *Science (New York, NY)* 324, 218-223.
- Kimura, M. (1964). Diffusion Models in Population Genetics. *Journal of Applied Probability* 1, 177-232.
- Kimura, M. (1983). *The Neutral Theory of Molecular Evolution* (Cambridge: Cambridge University Press).
- Korolev, K.S., Xavier, J.B., and Gore, J. (2014). Turning ecology and evolution against cancer. *Nat Rev Cancer* 14, 371-380.
- Krick, T., Verstraete, N., Alonso, L.G., Shub, D.A., Ferreira, D.U., Shub, M., and Sanchez, I.E. (2014). Amino Acid metabolism conflicts with protein diversity. *Mol Biol Evol* 31, 2905-2912.
- Lehninger, A., Nelson, D., and Cox, M. (2008). *Lehninger Principles of Biochemistry* (W. H. Freeman).

Lindqvist, L.M., Tandoc, K., Topisirovic, I., and Furic, L. (2018). Cross-talk between protein synthesis, energy metabolism and autophagy in cancer. *Current opinion in genetics & development* 48, 104-111.

Loayza-Puch, F., Rooijers, K., Buil, L.C.M., Zijlstra, J., F. Oude Vrielink, J., Lopes, R., Ugalde, A.P., van Breugel, P., Hofland, I., Wesseling, J., *et al.* (2016). Tumour-specific proline vulnerability uncovered by differential ribosome codon reading. *Nature* 530, 490-494.

McGranahan, N., and Swanton, C. (2017). Clonal Heterogeneity and Tumor Evolution: Past, Present, and the Future. *Cell* 168, 613-628.

Merlo, L.M.F., Pepper, J.W., Reid, B.J., and Maley, C.C. (2006). Cancer as an evolutionary and ecological process. *Nat Rev Cancer* 6, 924-935.

Nowell, P.C. (1976). The clonal evolution of tumor cell populations. *Science (New York, NY)* 194, 23.

Pavlova, N.N., and Thompson, C.B. (2016). The Emerging Hallmarks of Cancer Metabolism. *Cell metabolism* 23, 27-47.

Payne, S.H., and Loomis, W.F. (2006). Retention and Loss of Amino Acid Biosynthetic Pathways Based on Analysis of Whole-Genome Sequences. *Eukaryotic Cell* 5, 272-276.

Raiford, D.W., Heizer, E.M., Miller, R.V., Akashi, H., Raymer, M.L., and Krane, D.E. (2008). Do Amino Acid Biosynthetic Costs Constrain Protein Evolution in *Saccharomyces cerevisiae*? *J Mol Evol* 67, 621-630.

Riaz, N., Havel, J.J., Makarov, V., Desrichard, A., Urba, W.J., Sims, J.S., Hodi, F.S., Martin-Algarra, S., Mandal, R., Sharfman, W.H., *et al.* (2017). Tumor and Microenvironment Evolution during Immunotherapy with Nivolumab. *Cell* 171, 934-949.e915.

Schelker, M., Feau, S., Du, J., Ranu, N., Klipp, E., MacBeath, G., Schoeberl, B., and Raue, A. (2017). Estimation of immune cell content in tumour tissue using single-cell RNA-seq data. *Nature communications* 8, 2032.

Simińska, E., and Koba, M. (2016). Amino acid profiling as a method of discovering biomarkers for early diagnosis of cancer. *Amino Acids* 48, 1339-1345.

Svensson, V., Natarajan, K.N., Ly, L.H., Miragaia, R.J., Labalette, C., Macaulay, I.C., Cvejic, A., and Teichmann, S.A. (2017). Power analysis of single-cell RNA-sequencing experiments. *Nature methods* 14, 381-387.

Swire, J. (2007). Selection on Synthesis Cost Affects Interprotein Amino Acid Usage in All Three Domains of Life. *Journal of Molecular Evolution* 64, 558-571.

Tirosh, I., Izar, B., Prakadan, S.M., Wadsworth, M.H., 2nd, Treacy, D., Trombetta, J.J., Rotem, A., Rodman, C., Lian, C., Murphy, G., *et al.* (2016). Dissecting the multicellular ecosystem of metastatic melanoma by single-cell RNA-seq. *Science (New York, NY)* 352, 189-196.

Tsun, Z.-Y., and Possemato, R. (2015). Amino acid management in cancer. *Seminars in Cell & Developmental Biology* 43, 22-32.

Van Allen, E.M., Wagle, N., Stojanov, P., Perrin, D.L., Cibulskis, K., Marlow, S., Jane-Valbuena, J., Friedrich, D.C., Kryukov, G., Carter, S.L., *et al.* (2014). Whole-exome sequencing and clinical interpretation of formalin-fixed, paraffin-embedded tumor samples to guide precision cancer medicine. *Nature medicine* 20, 682-688.

Vander Heiden, M.G., and DeBerardinis, R.J. (2017). Understanding the Intersections between Metabolism and Cancer Biology. *Cell* 168, 657-669.

Wagner, A. (2005). Energy Constraints on the Evolution of Gene Expression. *Molecular Biology and Evolution* 22, 1365-1374.

Waldman, Y.Y., Geiger, T., and Ruppin, E. (2013). A genome-wide systematic analysis reveals different and predictive proliferation expression signatures of cancerous vs. non-cancerous cells. *PLoS genetics* 9, e1003806.

Wu, C.-I., Wang, H.-Y., Ling, S., and Lu, X. (2016). The Ecology and Evolution of Cancer—The Ultra-Microevolutionary Process. *Annual review of genetics*.

REVIEWERS' COMMENTS:

Reviewer #2 (Remarks to the Author):

The authors has provided a satisfactory response to my previous comments. Specifically, they have demonstrated that the amino acids cost index has a predictive power beyond cell proliferation. The authors have also added data relating the amino acids cost index with the concentrations of circulating amino acids. This data helps to understand the authors thesis that during tumor progression the utilization of amino acids is optimized. Overall this is an important contribution to our understanding of the evolutionary pressures acting on cancer metabolism.

Reviewer #3 (Remarks to the Author):

All my previous concerns have been addressed.

Reviewer #4 (Remarks to the Author):

Authors have made significant improvements in response to previous concerns. However, the actual meaning/definition of the ECPA metric still not entirely clear and should be better described. This is a key point as it the basis for the entire manuscript. Previous concerns about the relatively small magnitude of effect and limited clinical utility stand. Fig1F regression is skewed, seems to bias towards outliers on right of plot?

Response to reviewers' comment

Reviewer #2 (Remarks to the Author):

The authors has provided a satisfactory response to my previous comments. Specifically, they have demonstrated that the amino acids cost index has a predictive power beyond cell proliferation. The authors have also added data relating the amino acids cost index with the concentrations of circulating amino acids. This data helps to understand the authors thesis that during tumor progression the utilization of amino acids is optimized. Overall this is an important contribution to our understanding of the evolutionary pressures acting on cancer metabolism.

Response: This reviewer well summarized the changes we made in our previous revision. We thank this reviewer for the supports.

Reviewer #3 (Remarks to the Author):

All my previous concerns have been addressed.

Response: We thank the reviewer for the positive feedbacks.

Reviewer #4 (Remarks to the Author):

Authors have made significant improvements in response to previous concerns. However, the actual meaning/definition of the ECPA metric still not entirely clear and should be better described. This is a key point as it the basis for the entire manuscript.

Response: Thanks for the suggestion. In this revision, we added the sentence “Therefore, lower ECPA values indicate reduced relative usage of expensive AAs and vice versa.” in the introduction (Line 70-71, Page 4) to further clarify the meaning of ECPA.

Previous concerns about the relatively small magnitude of effect and limited clinical utility stand.

Response: Thanks for raising these concerns. In this revision, we have toned down the claims regarding the clinical utility. Our efforts are summarized as follows:

(1) We changed the title from “Decreased Biosynthetic Energy Cost for Amino Acids in Cancer Evolution and Its Potential Clinical Utility” to “Biosynthetic Energy Cost for Amino Acids Decreases in Cancer Evolution” (Line 1, Page 1).

(2) We also changed the sentence “ECPA not only shows robust prognostic power across many cancer types, but also improves the prediction of tumor response to checkpoint inhibitor immunotherapy. Our ECPA analysis

reveals a common principle during cancer evolution and demonstrates its potential clinical utility.” in the abstract of our previous submission to “Our ECPA analysis reveals a common principle during cancer evolution” Page 2).

(3) We removed the sentence “We show that our ECPA metrics are informative as both prognostic and predictive markers” from the Introduction (Line 72, Page 4).

(4) We removed the sentence “There has been growing interest in AA metabolism for cancer diagnosis and therapy^{16,32,95}, and our ECPA concept provides novel insights into such clinical practice” from Discussion of our previous submission (Line 364, Page 14).

We hope these changes are satisfactory to you.

Fig1F regression is skewed, seems to bias towards outliers on right of plot?

Response: We apologize for the confusion. In Figure 1f, we aimed to show that the Cost-Usage (C-U) anticorrelation in animals is weaker using the H11 metric (y-axis) compared to the Y20 metric (x-axis). The red line in Figure 1f means $y = x$ rather than a regression line. Our conclusion is supported by the fact that most dots are located above $y = x$. We realized that the regression analysis could not serve this purpose and performed a Wilcoxon signed-rank test instead in this revision. The legend for Figure 1f now becomes:

“C-U anticorrelation in animals is weaker using H11 metric compared to Y20 metric (Wilcoxon signed-rank test, $P = 3 \times 10^{-61}$). The red line indicates where $y = x$.” (Line 766-767, Page 27).